# Necroptosis in both tumour and stromal compartments determines responsiveness to immunogenic cell death-based immunotherapy

Immunotherapy has transformed cancer treatment, including early triple-negative breast cancer (TNBC), yet most patients with advanced TNBC fail to respond to immune checkpoint blockade (ICB) plus chemotherapy. Durable control likely requires not only tumour cell killing but also immunogenic cell death (ICD) that activates antitumour immunity. Using a $Brca1^{-/-}p53^{-/-}$ organoid-derived TNBC model that recapitulates the immune landscapes of basal-like tumours, we show that RIPK1-driven ICD synergises with anti-PD-1 therapy to induce durable tumour control and immune memory in immune-infiltrated tumours. Mechanistically, both tumour-intrinsic and stromal necroptosis are required. Deletion of *Ripk1* or *Mlkl* in tumour cells, or *Mlkl* in the stromal compartment, markedly impairs therapeutic efficacy. Moreover, immunologically "cold" tumours can be rendered responsive to ICD-based therapy by STING agonists. These findings demonstrate that the benefit of IAP antagonism with checkpoint blockade critically depends on coordinated necroptosis in both tumour and stromal cells, underscoring the need to integrate tumour microenvironmental context when designing ICD-targeted immunotherapies.

Breast cancer remains a major health threat worldwide, accounting for 30% of all female cancer cases and 15% of female cancer-related deaths[1]. Breast cancer exhibits a high degree of inter-patient heterogeneity not only between the different subtypes of breast cancer but also within them, necessitating a more personalised approach to therapy. Additionally, the various breast cancer subtypes differ significantly in their tumour microenvironments, with triple-negative breast cancers (TNBC) and HER2-positive types being generally considered more immune-active ("hot tumours"), while ER+ luminal carcinomas are typically immune-deficient ("cold tumours").

TNBCs are primarily defined by the lack of expression of the oestrogen receptor (ER), progesterone receptor (PR), and HER2 (ERBB2), which are typically present in other forms of the disease. Although there have been recent advancements in the targeted treatment of TNBC,

including the use of PARP inhibitors in those with BRCA1/2 mutation or immunotherapy with anti-PD-1 or PD-L1 agents, many patients still do not respond to these treatments or develop resistance. Although immunotherapy has revolutionised anti-cancer therapies, treatment with immune checkpoint inhibitors (ICI) has limited efficiency in the majority of patients. Accordingly, single-agent ICI activity has been observed in <10% of patients with metastatic disease[2–4]. Moreover, it is difficult to predict who might respond as current biomarkers of ICI have several limitations[5]. Further, tumours may evolve resistance to immunotherapies, for example due to a lack of immunogenic tumour antigens, changes in cancer cell signalling pathways, the presence of inhibitory checkpoint molecules, and a 'cold' tumour microenvironment (TME) characterised by poor recruitment of tumour-infiltrating lymphocytes (TIL) and antigen presentation, along with immune-

✉e-mail: jarama.clucas@icr.ac.uk; tencho.tenev@icr.ac.uk; pascal.meier@icr.ac.uk

suppressive factors like myeloid-derived suppressor cells (MDSCs), M2-like macrophages, and the presence of regulatory T cells (Tregs)[6–8].

Current immunotherapy approaches in breast cancer aim to address T-cell priming issues by combining chemotherapy with ICIs. Accordingly, Pembrolizumab, in combination with chemotherapy, has been incorporated into the standard-of-care treatment for patients with early TNBC, and both Pembrolizumab and Atezolizumab in patients with PD-L1-positive early or advanced TNBC[9,10]. The death of cancer cells plays a crucial role in the cancer immunity cycle[11,12], as cancer cell death is required for efficient activation of the immune system. While induction of cancer cell death is critical for the uptake of tumour-associated antigens by antigen presenting cells, the death of cancer cells alone is not sufficient for the cross-priming of CD8+ T cells. In addition, efferocytosing antigen presenting cells, such as immature dendritic cells (DCs), must also be exposed to adjuvants that trigger their maturation[13]. Importantly, not every cell death modality is capable of providing the necessary immunological signals that can skew the maturation process of peripheral DCs towards an immunogenic fate. While apoptosis is the most prominent cell death pathway, apoptotic execution tends to be immunologically silent because the fast dismantling and clearance of the dying cell minimises the release of immunogenic signals[14]. Apoptotic caspases also cleave and incapacitate signalling molecules, such as cyclic GMP-AMP synthase (cGAS), Interferon regulatory factor 3 (IRF3) and receptor-interacting serine/threonine-protein kinase 1 (RIPK1), that would otherwise drive the production of immunogenic danger signals[10,15]. By contrast, cells that die by caspase-independent forms of cell death, such as necroptosis, release 'alarmins', which alert the immune system of danger[11,12].

Necroptosis, which is an explosive form of cell death, is the body's natural method of killing cells infected by pathogens. Cancer cells that die in this way provide both antigens and adjuvants for DCs, which in turn activate CD8+ T cells through antigen cross-priming[14,16]. Therefore, developing therapies that drive immunogenic necroptosis of breast cancer cells could (re)activate the immune system of a patient to attack cancer cells. Necroptosis is typically triggered by the activation of RIPK1, RIPK3, and mixed lineage kinase domain-like pseudokinase (MLKL), which ultimately facilitates the release of these inflammatory signals by causing plasma membrane rupture[17–20]. RIPK1 acts as a key stress sentinel, that operates downstream of many cytokine receptors and pattern recognition receptors (PRRs). RIPK1 can simultaneously coordinate transcriptional inflammatory gene expression with different cell death pathways, including immunogenic apoptosis, necroptosis and pyroptosis[14,21–23].

Since RIPK1-mediated cell death is a highly immunogenic process, it is subject to tight regulatory control. Members of the Inhibitor of Apoptosis (IAP) protein family, such as cIAP1, cIAP2 and XIAP function as pivotal gatekeepers of RIPK1 and RIPK1-mediated complexes, such as the ripoptosome and necrosome, which mediate immunogenic cell death[14,16,24,25]. The three clinically relevant IAPs, cIAP1, cIAP2 and XIAP are frequently upregulated at the transcript and protein levels in breast cancer and contribute to treatment resistance and poor clinical outcomes[26–31]. cIAP1/2, and to some extent XIAP, modulate ubiquitin-dependent pathways that prevents assembly of complex II/Ripoptosome/Necrosome—a cytosolic signalling platform responsible for the integration of cellular stress, innate immune signalling pathways as well as cell death[14,31–35]. In parallel, IAPs also regulate NFκB and interferon-mediated production of inducible damage-associated molecular patterns (iDAMPs) that can stimulate recruitment and activation of DCs[36,37], leading to efficient cross-priming of effector T cells[16]. Additionally, XIAP can effectively inhibit caspases 3, 7 and 9 by directly binding to their active sites[34]. To promote cell death, drugs that mimic natural IAP antagonists, such as second mitochondria-derived activator of caspases (Smac/DIABLO, frequently referred to as SMAC mimetics or IAP antagonists) were developed. Such IAP antagonists mediate most

of their effects via acute activation of RIPK1, highlighting their potential to enhance anti-tumour responses through the modulation of RIPK1-driven apoptosis and necroptosis. Several IAP antagonists have entered clinical trials. The third generation IAP antagonist ASTX660 (Tolinanant), which is currently undergoing clinical testing[38–40], offers balanced inhibition of cIAP1/2 and XIAP[38,41]. This is important as all three IAPs regulate RIPK1 in a redundant manner.

Activation of immunogenic forms of cell death has emerged as an attractive approach to stimulate an anti-tumour immune response. However, developing therapies that complement ICB is hindered by the use of immunocompromised mice in preclinical studies. Additionally, there is a distinct lack of suitable genetically engineered mouse models (GEMMs) capable of producing diverse, invasive TNBC, including the majority of basal-like breast carcinomas, with varying immune environments that reflect the human disease. Using a tumour organoid-based transplantation model in immunocompetent mice, which reliably mimics the range of tumour immune microenvironments (TIME) of human TNBC with short latency and a stable tumour micro-environment across passages, we have evaluated the response of ICI-resistant tumours to combination therapies that drive lytic forms of cell death. We demonstrate that ICI-resistant and immune-escaped tumours respond to RIPK1-driven immunogenic cell death in combination with anti-PD-1 therapy, particularly under necroptotic conditions, with increased recruitment of CD8+ T cells and the development of immune memory. Importantly, we find that the efficacy of IAP antagonism with ICB critically depends on coordinated necroptosis in both tumour and stromal compartments to amplify immune activation, emphasising that the therapeutic outcome of ICD-based immunotherapies is dictated by the cellular architecture and immune composition of the tumour microenvironment. Immunologically colder tumours, that do not respond to this treatment combination, can be sensitised to RIPK1-mediated immunogenic cell death through pathogen mimicry, such as STING agonism. Together, these findings indicate that the combination of RIPK1 activation with targeted immune-modulating therapies has the potential to overcome ICI resistance and enhance outcomes for patients with TNBC. However, it will be important to stratify patients based on their cellular composition and immune landscape of the tumour microenvironment.

## Results

### Brca1p53 deficient tumours display heterogeneous architecture and immune landscape

To study how to boost anti-tumour immunity in TNBC, we developed a transplantation-based syngeneic mouse cancer model that faithfully and stably recapitulates the phenotypic diversity of the human disease. To achieve this, we utilised the *Blg-Cre;Brca1[f/f]p53[+/−]* mouse model[42,43], subsequently referred to as BP. In this model, tumours develop stochastically over 8 to 16 months due to the deletion of the *Brca1* gene and subsequent loss of p53 heterozygosity in mammary epithelial luminal progenitor (Fig. 1A–C)[42,43]. Histopathological analysis and multiplex immunofluorescent imaging demonstrated that *Blg-Cre;Brca1[f/f]p53[+/−]* females present with invasive carcinomas, most of no special type (NST)[43]. Although the same tumour suppressors were lost in the luminal progenitor cells of the mammary gland in every animal (Fig. 1B and Supplementary Fig. 1A, B), tumours from different animals showed differences in nuclear morphology and tissue microarchitecture, including tumour/stroma ratio, demonstrating substantial inter-tumour heterogeneity (Fig. 1D and Supplementary Fig. 1C). To facilitate pre-clinical experimentation, we derived primary tumour organoid lines (O) from each tumour (Fig. 1C).

### Organoid-derived tumours faithfully recapitulate key features of their primary tumour of origin

We selected the organoid lines BP903, BP487 and BP962 for further characterisation as they were derived from tumours with distinct

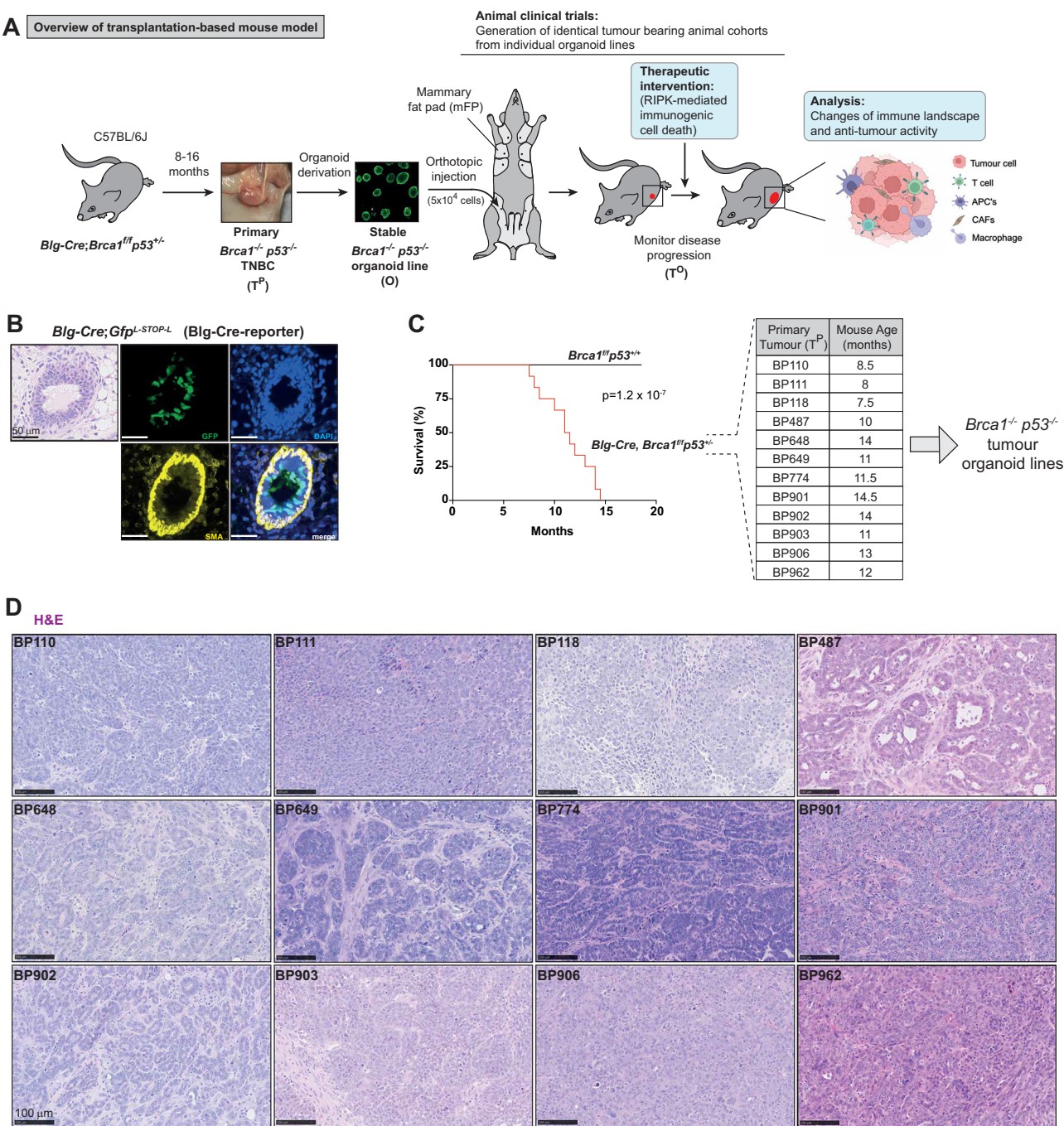

**Fig. 1 | Development of a transplantation-based mouse model of TNBC.**
**A** Schematic representation of the tumour-organoid pipeline, enabling us to generate large mouse patient cohorts for the development of treatment approaches. Organoids (O) were generated from spontaneously forming primary tumours (T$^P$s) occurring in *Blg-Cre;Brca1$^{f/f}$p53$^{+/-}$* females. To establish large groups of mice with identical tumours, organoids from a single primary tumour were orthotopically injected into the mammary fat pads of naïve mice. As tumours develop, these tumour organoids (T$^O$s) maintain complex tumour microenvironments, allowing the study of immunogenic cell death. **B** Representative H&E and confocal images of Cre-reporter expression in ducts within the mammary fat pad of *Blg-Cre;Gfp$^{L-STOP-L}$* mice ($n = 3$ animals). Cre-reporter expression was observed in mammary luminal epithelial cells (Green), Myoepithelial cells labelled with SMA (Yellow), Nuclei labelled with DAPI (Blue). Scale bar depicts 50 μm. **C** Kaplan−Meier Survival curves of *Blg-Cre;Brca1$^{f/f}$p53$^{+/+}$* mice ($n = 13$) versus *Blg-Cre;Brca1$^{f/f}$p53$^{+/-}$* ($n = 12$). Log-rank (Mantel−Cox) test. Table indicating the tumour line and the corresponding age of the mouse in which the tumour formed. **D** Representative H&E images of the respective T$^P$. H&E staining was performed on a single primary tumour per mouse (**A**). Scale bar 100 μm. Source data are provided as a Source Data file and in Zenodo: https://doi.org/10.5281/zenodo.18130193.

histopathological features (Fig. 1D and Supplementary Fig. 1C). The organoids derived from the primary BP903, BP487 and BP962 tumours retained their tumorigenic capacity as their orthotopic transplantation into naïve, immune competent C57BL/6J females gave rise to organoid-derived tumours (T$^O$) within a 30−40 period (Supplementary Fig. 2A). The respective primary organoids exhibited significant differences from each other (Fig. 2A). BP487 organoids formed highly organised and polarised structures with branching buds and hollow lumens (Fig. 2A). In addition, BP487 organoids had prominent E-cadherin localisation at cell-cell contacts, which is highly reminiscent to organoids derived from normal, healthy mammary glands. BP487 organoids were comparatively the

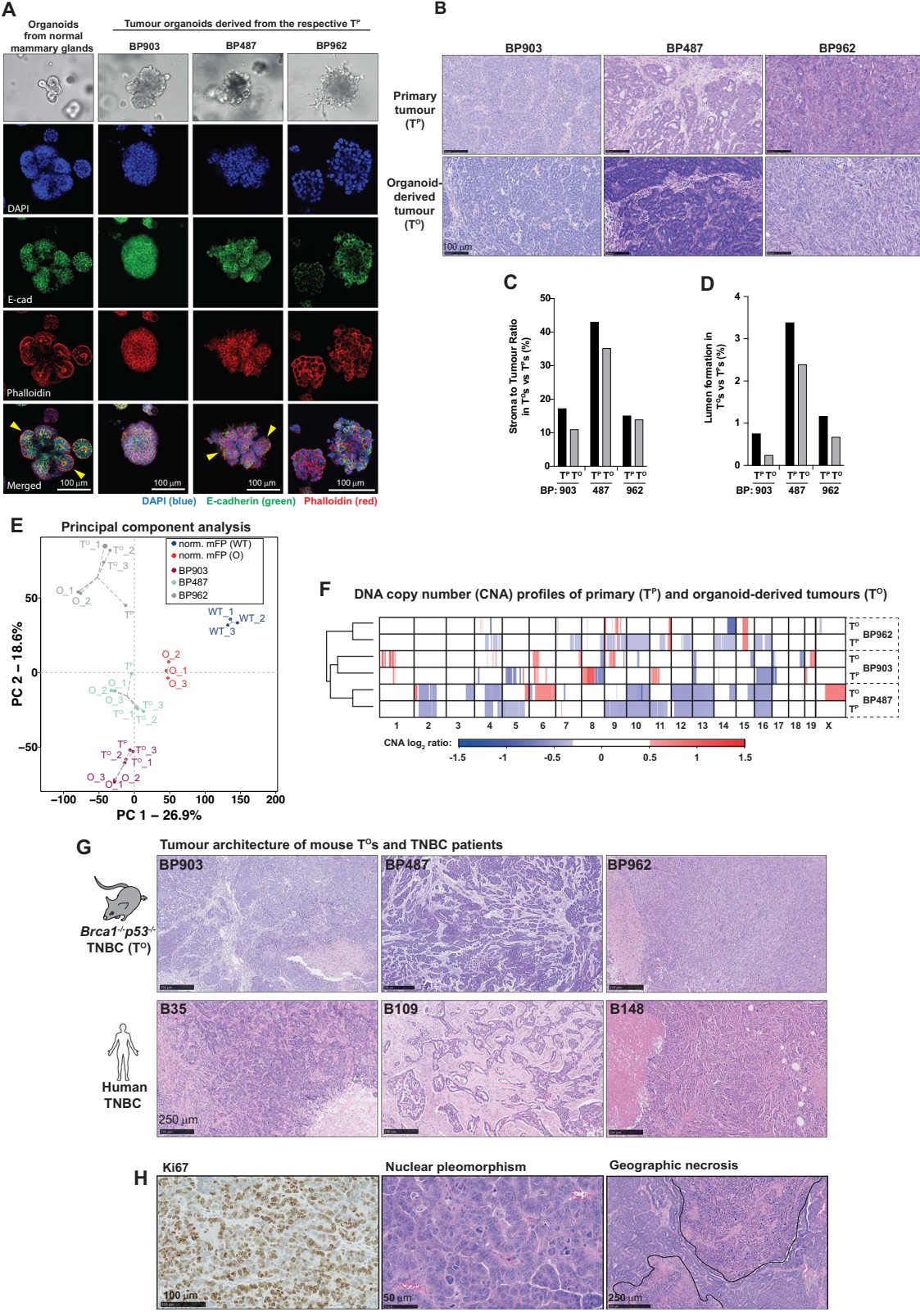

most differentiated. In contrast, BP903 and BP962 organoids lacked polarity and did not form lumens. While BP903 organoids retained E-cadherin expression, localisation appeared to be abnormal, with strong staining within the cytoplasm as well as at cell membranes (Fig. 2A). BP962 organoids displayed two distinct populations of epithelioid and mesenchymal-like, spindle-shaped cells (Fig. 2A), which were also observed in vivo in the primary BP962 tumour (T$^P$)

and organoid-derived tumours (T$^O$) (Fig. 2B). While the epithelioid cells were E-cadherin positive, the mesenchymal-like, spindle-shaped cells were negative for E-cadherin expression (Supplementary Fig. 2B). The spindle cells surrounded the periphery of the epithelioid clusters, protruding into the matrix in vitro (Fig. 2A). These mesenchymal-like subpopulations remained present over passaging, indicating that the BP962 line produces epithelial and metaplastic

**Fig. 2 | Organoid derived tumours faithfully recapitulate their corresponding primary tumour. A** Representative brightfield and confocal images from three independent experiments illustrating the distinct morphological characteristics of the indicated organoid lines, derived from normal mammary fat pads and primary tumours (T$^P$s), respectively. Yellow arrows depict branches and polarised buds. Nuclei (Blue), Phalloidin (Red), E-cadherin (Green). Both Brightfield and confocal images were captured at 20x objective. Scale bar 100 μm. **B** Representative H&E images of the indicated T$^P$s and their corresponding T$^O$s. H&E staining was performed on T$^P$ (see 1A) and T$^O$s, T$^P$ sections were generated from a single parental tumour, whereas T$^O$s were derived from more than three independent animals. Scale bar 100 μm. **C** Quantification showing the stroma-to-tumour ratio in primary tumours (T$^P$s) and their corresponding tumour organoids (T$^O$s). Values were calculated from a single central section per tumour ($n = 1$ per T$^P$s and corresponding T$^O$s). **D** Quantification showing lumen formation in T$^P$s and their corresponding T$^O$s. Values were calculated from a single central section per tumour ($n = 1$ per T$^P$s and

corresponding T$^O$s). **E** Principal component analysis (PCA) showing distinct clustering of T$^P$s, their corresponding organoids (Os), and the subsequent T$^O$s. Each data point represents an individual sample. **F** Unsupervised hierarchical clustering based on DNA copy number (CNA) profiles of T$^P$s and corresponding T$^O$s. T$^P$ sections were generated from a single parental tumour, whereas T$^O$s were derived from three independent animals. **G** Representative H&E images showing the tumour architecture of murine tumour organoids (T$^O$s) and human TNBC samples. T$^O$s were derived from more than three independent animals, and each human TNBC sample represents a tumour from an individual patient. Scale bar: 250 μm. **H** Representative IHC images illustrating typical histopathological features of basal-like TNBC samples, including proliferation (Ki67), nuclear pleomorphism and geographical necrosis present in BP487 T$^O$s and T$^P$s (Supplementary Fig. 2C). Images depict characteristic features used for pathological assessment across three independent primary tumours. Scale bar: 100 μm, 50 μm and 250 μm. Source data are provided as a Source Data file and in Zenodo: https://doi.org/10.5281/zenodo.18130193.

cancer cell populations, a feature that is often associated with the cellular programme of epithelial-mesenchymal transition (EMT) and confers drug resistance, metastatic capacity and poor prognosis in TNBC[44,45].

Importantly, T$^O$s retained the histological features of their parental tumour (T$^P$) from which the organoid line was derived, faithfully recapitulating morphological features and stromal composition (Fig. 2B–D). BP487 T$^O$s displayed the highest stroma to tumour ratio, and BP487 cancer cells were arranged in duct-like structures. Notably, the individual organoid lines remained stable and reproducible across passages, as evidenced by mRNA and mutational profiling, while some clonal and CNA differences relative to the parental tumours were observed, reflecting tumour heterogeneity and adaptation in culture (Fig. 2E, F). Furthermore, the murine BP tumours exhibited multiple cytological and architectural features typically seen in human basal-like breast cancer (Fig. 2G)[42,43,46], including high mitotic count, nuclear pleomorphism, high nuclear-cytoplasmic ratio, and geographic necrosis (Fig. 2H and Supplementary Fig. 2C), which is consistent with a previous report[47]. mRNA abundance-based Absolute Assignment of Breast Cancer Intrinsic Molecular Subtype (AIMS) analysis demonstrated that the BP lines retained their basal-like features in vitro and in vivo (Supplementary Fig. 2D). Additionally, T$^P$s and T$^O$s of all three BP lines expressed CK14, a basal marker for myoepithelial cells in the mouse mammary gland, but which is commonly expressed in luminal progenitor-origin tumours acquiring a basal-like phenotype[48] (Supplementary Fig. 2E, F). While BP487 and BP962 were negative for oestrogen receptor (ER), progesterone receptor (PR) and human epidermal growth factor receptor 2 (Her2), BP903 showed weak but detectable expression for ER and PR staining (Supplementary Fig. 2E–G). This finding aligns with previous reports, indicating that approximately 19% of spontaneous tumours in *Blg-Cre;Brca1$^{f/f}$p53$^{+/-}$* mice are ER-positive[43], while 10–36% of *BRCA1* mutant patients present with ER$^+$ tumours[49]. Despite some BP903 cells being ER$^+$, ER receptor activity appeared not to be required for cell survival, as ER inhibition with fulvestrant and tamoxifen had no impact on organoid viability (Supplementary Fig. 2H).

### High diversity in immune landscape of *Brca1$^{-/-}$p53$^{-/-}$* tumours is captured in organoid-derived tumours

BP903, BP487 and BP962 T$^O$s had distinctly different profiles of infiltrating lymphocytes and myeloid cells, with BP903 tumours showing the lowest level of lymphocyte infiltration (Fig. 3A, B and Supplementary Fig. 3A). BP487 T$^O$s had high infiltration of monocytes, particularly of the Ly6c$^+$ subset, suggesting a 'pro-inflammatory' monocytic tumour immune microenvironment. In contrast to BP903 and BP487 tumours, BP962 T$^O$s were highly infiltrated by dendritic cells (DCs), CD4$^+$ and CD8$^+$ T cells, B cells, neutrophils and macrophages. In silico deconvolution of cell types using CIBERSORTx indicated that BP962 tumours are rich in myeloid cells (Fig. 3C), which was confirmed using

IHC analysis (Fig. 3D, E). Compared to BP903 and BP487, T$^O$s from BP962 had greater frequency of F4/80$^+$ cells (macrophages), that were sustained during tumour growth (Supplementary Fig. 3B). A high abundance of tumour-associated macrophages (TAMs) was recently reported in TNBC patients, accounting for 50% of all cells of the tumour micro-environment (TME)[50–52]. Therefore, BP962 tumours may replicate a comparable immune environment to that observed in human disease. The TIL profiles of T$^O$s were highly similar to their respective T$^P$s, indicating that the organoids were capable of 'recreating' a tumour micro-environment that closely resembled the tumour from which they were derived. Gene set enrichment analysis (GSEA) showed significant enrichment of macrophage-specific transcripts in BP962 T$^O$s, compared to BP903 and BP487 T$^O$s (Fig. 3F), whereas BP903 and BP487 T$^O$s had similar levels to each other (Supplementary Fig. 3C).

Although the microenvironment of BP962 tumours exhibited substantial infiltration of pro-inflammatory immune cells, and appeared immunologically 'hot', the tumour cells had significantly increased expression of immune checkpoint proteins, including PD-L1 and T cell immunoreceptor with Ig and ITIM domains (TIGIT), when compared to other BP lines (Fig. 3G–I and Supplementary Fig. 3D). Furthermore, BP962 tumours showed a notably downregulated expression of genes associated with the MHC class I antigen presentation pathway (Supplementary Fig. 3E, F). Nevertheless, compared to BP903 and BP487, BP962 T$^O$s were rich in the expression of NF-κB- and IFN-dependent target genes (Fig. 3J), which is frequently associated with aberrant cell survival, cell cycle progression, inflammation, metastasis, angiogenesis and regulatory T cell function[53]. Notably, such NF-kB- and IFN signatures are also known to correlate with improved responsiveness to immune-checkpoint blockade-based therapies as both these signalling pathways influence bystander killing of tumour cells[54,55]. Accordingly, elevated NF-kB and IFN pathways correlate with improved responses to necroptosis. To address the role of the immune compartment on tumour growth, we inoculated immunocompetent C57BL/6J or immunocompromised (NSG) females with the three organoid lines. BP962-derived tumours showed a more aggressive phenotype when grown in NSG mice in comparison with immunocompetent C57BL/6J mice (Supplementary Fig. 4A). In contrast, BP487 and BP903 tumours grew at the same rate in NSG and C57BL/6J animals, suggesting that these tumours are poorly infiltrated by CD8+ T cells and do not actively engage host immunity that contributes to tumour control (Fig. 3A–C). Further, BP962 tumour-bearing C57BL/6J animals presented with enlarged spleens, when compared to controls, NSG mice or mice bearing BP903 tumours, suggesting that this secondary lymphoid organ participates in restraining the growth of BP962 cancer cells (Supplementary Fig. 4B, C). Intriguingly, we noticed that BP962 organoids and organoid-derived tumours produced high levels of cytokines such as CSF1 (Supplementary Fig. 4D), which can drive inflammation and immune cell proliferation. This cytokine production

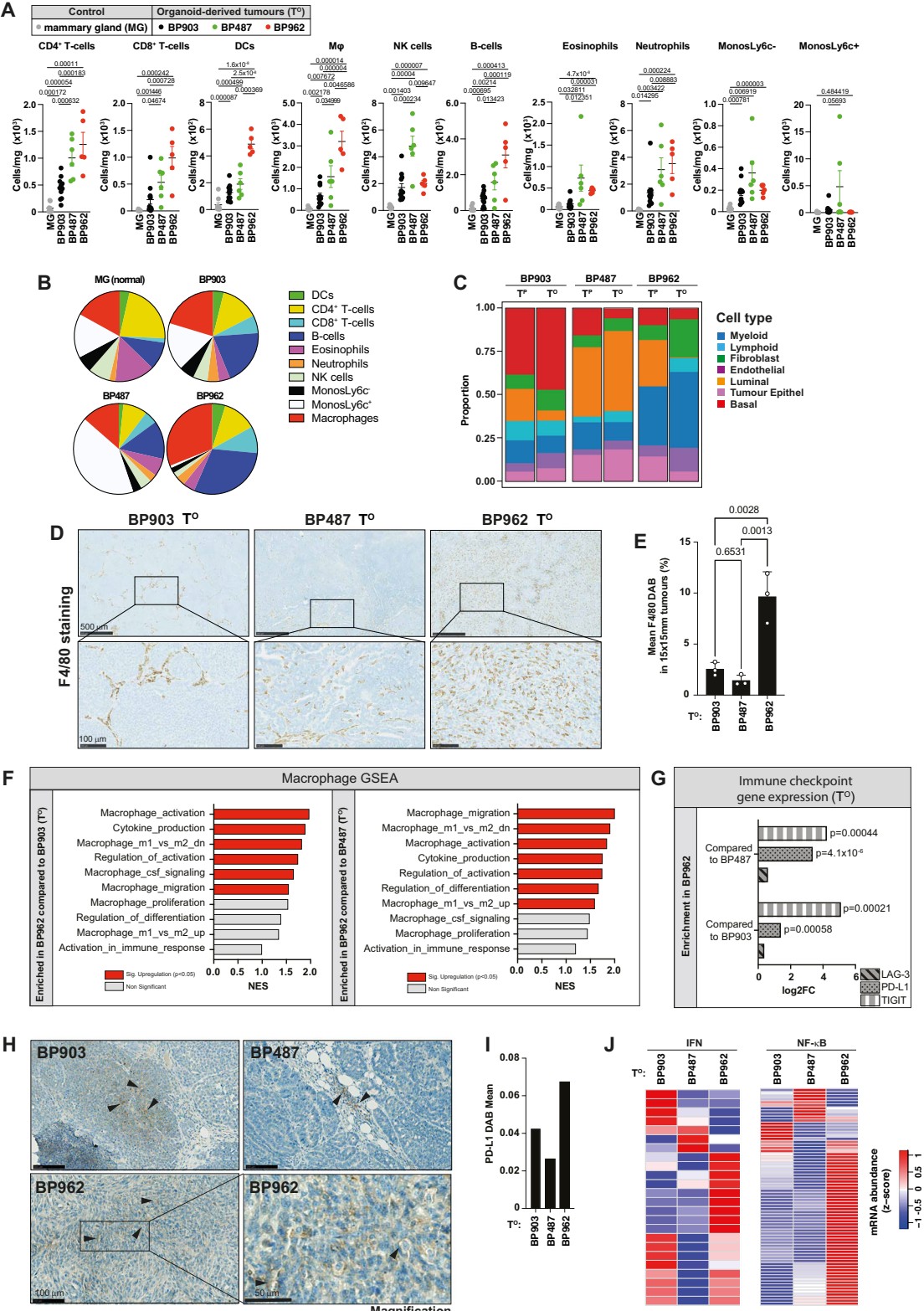

can lead to systemic inflammation, which causes the spleen to enlarge as it processes an increased load of immune cells and inflammatory signals[56]. However, additional tumour-related mechanisms may also contribute to spleen enlargement. Although host animals responded immunologically to the presence of BP962 cancer cells, BP962 tumours ultimately escape immune control. Therefore, BP962 tumours serve as an ideal model to study immune-escaped tumours,

which represent a particular challenge for cancer immunotherapy[57]. Taken together, these data demonstrate that the BP model gives rise to tumours with very different immune landscapes. The high diversity in the immune environment of the different BP lines is comparable to that observed in the human disease of high-grade HR-deficient TNBC[58]. Whilst T[O]s generated from the different lines are distinct with respect to their tumour/stroma ratio and immune composition, the T[O]s are

**Fig. 3 | T°s exhibit diverse immune landscape profiles. A** Flow cytometric analysis (FACS) of the indicated mammary gland and T°s tumours. Mammary fat pads (MG), BP903 (Black), BP487 (Green), and BP962 (Red). Each dot represents a single sample from one animal, either a mammary fat pad or a tumour. See Supplementary Fig. 3 for gating strategy. Data show mean ± SEM. **B** Pie charts illustrating the distribution of the indicated immune cell subtypes based on the FACS analysis shown in (**A**). **C** Stacked bar chart depicting the relative fractions of cell type composition in one primary T°s and three corresponding T°s, as determined by CIBERSORTx deconvolution of RNA-seq data. **D** Representative IHC images showing F4/80 positive cells (brown) in the indicated T°s. The top panels show a wide-field view (scale bar: 500 μm), while the bottom panels display a zoomed-in view of the indicated areas (scale bar: 100 μm). Images are representative of three independent T°s. **E** Quantification of F4/80 positivity in the indicated T°s. Each dot represents an individual tumour. **F** Gene set enrichment analysis (GSEA) of the indicated macrophage gene sets in BP962 T°s compared to BP903 T°s (left panel) and BP487 T°s (right panel). Colours indicate upregulated pathways: red for significant (*p* < 0.05) and grey for non-significant pathways. See Supplementary Data 2 for further information. Data are derived from three independent T°s per group.

**G** Differential analysis of the indicated immune checkpoint genes, based on RNA expression in BP962 T°s versus BP487 T°s (top) and BP962 T°s versus BP903 T°s (bottom). Data are derived from three independent T°s per group. **H** IHC staining showing PD-L1 expression (brown) in the indicated T°s. Images are representative of a section from a single tumour. **I** Quantification of PD-L1 expression based on DAB mean intensity values derived from the images shown in (**H**) across the indicated T°s. **J** Heat maps from bulk RNA-seq displaying the relative mRNA abundance of IFN signalling genes (left) and NFκB genes (right) in the indicated T°s. Gene names are provided in Supplementary Data 3. Data are derived from three independent T°s per group. Data show mean ± SD. *p* values were calculated using multiple two-sided unpaired *t*-test (**A**) or one-way ANOVA (Tukey's multiple comparison test) (**E**). The enrichment score (**F**) was obtained using the Kolmogorov–Smirnov statistic and the *p* value using the permutation test. Statistical testing for RNA-Seq data (**G**) was performed using quasi-likelihood negative binomial generalised log-linear model (function glmQLFTest from edgeR). Source data are provided as a Source Data file and in Zenodo: https://doi.org/10.5281/zenodo.18130193.

highly reproducible within each line, enabling the generation of large mouse 'patient' cohorts, in which to evaluate drug combinations aimed at triggering immunogenic cell death and driving more effective anti-tumour immune responses.

## Targeting immune-escaped tumours by RIPK1-mediated cell death

Although the immune system can potentially be harnessed to drive anti-tumour responses, tumour evolution favours the survival of poorly immunogenic tumour clones and creates a microenvironment that suppresses effective anti-tumour immunity[59]. However, by inducing immunogenic cell death, it may be possible to counteract these evasion strategies, thereby enhancing the immune system's ability to recognise and eliminate tumour cells, and thus improving therapeutic outcomes[24,25,57,60–63]. To test whether it is possible to induce an anti-tumour immune response by targeting immunogenic cell death, we focussed on Receptor-interacting serine/threonine-protein kinase 1 (RIPK1), which functions as a critical stress sentinel, coordinating cellular survival, inflammatory responses and immunogenic cell death in the form of apoptosis, necroptosis and pyroptosis[14,21–23]. Activation of RIPK1 is tightly regulated by members of the IAP protein family[24,64–66], albeit RIPK1 is additionally regulated by further checkpoints[24,67,68], including caspase-8 mediated cleavage[69–72] (Fig. 4A). Small molecule IAP antagonists, such as ASTX660 (tolinapant), have been developed and are undergoing phase I clinical testing in combination with pembrolizumab (MK-3475) (ASTEROID, NCT05082259). While IAP antagonists were initially developed to promote tumour cell apoptosis, IAP antagonists also induce anti-tumour activity through modulation of innate and adaptive immunity, rejuvenating exhausted immune cells and driving immunogenic forms of cell death (Fig. 4A).

While RIPK1 was barely detectable in normal mammary glands, its expression was markedly elevated in cancer cells of primary BP tumours (Fig. 4B and Supplementary Fig. 5A), with BP962 cancer cells being among the highest RIPK1 expressers (Supplementary Fig. 5B). Intriguingly, IHC-based H-score quantification of RIPK1 protein expression specifically in cancer cells indicated that its expression was not uniform in cancer cells but varied across tumour cells of each BP line as well as between lines. In contrast to BP962 tumours, cancer cells from primary BP487 tumours had the lowest RIPK1 expression (H score of 5), which was comparable to the levels of RIPK1 in cells of normal mammary glands, including in the epithelial cells of the ducts. BP903 tumours had a RIPK1 H score of 115. The differences in RIPK1 expression between the different BP lines were maintained in their respective primary organoids (O) and T°s (Fig. 4C, D). Although the different BP organoid lines expressed different levels of components of the RIPK1 cell death signalling cascade (RIPK1, RIPK3, caspase-8 and MLKL), the

BP903, BP487, and BP962 organoids, like E0771 TNBC-derived spheroids, readily died following treatment with necroptotic stimuli such as TNF/SM/emricasan (TNF/SM/E) and SM/E, whereby emricasan (E) represents a pan-caspase inhibitor (Fig. 4E, F and Supplementary Fig. 5C–H). Treatment with either TNF, ASTX660 (IAP antagonist), emricasan or TNF/ASTX660 had no or little effect on organoid viability. Cell death upon treatment with SM/E and T/SM/E was suppressed by co-treatment with inhibitors of RIPK1, RIPK3 or genetic deletion of *Mlkl* (Fig. 4E, G and Supplementary Fig. 5D), indicating that IAP antagonism sensitises cells to RIPK1-mediated necroptosis. While normal mammary organoids were resistant to necroptosis (Fig. 4H), BP tumour organoids and E0771 TNBC spheroids were susceptible to this lytic form of cell death (Fig. 4E, I). Induction of necroptosis with two chemically distinct IAP antagonists, ASTX660 and SM164, produced comparable results (Fig. 4E and Supplementary Fig. 5D), providing confidence that this effect is driven by an on-target mechanism.

## Boosting immunotherapy by targeting IAPs

To test whether IAP blockade may help to maximise immunogenic cell death and the generation of an efficient adaptive immune response, we conducted co-clinical studies in immunocompetent C57BL/6J mice that resemble the treatment protocol employed in the ASTEROID phase I clinical trial (NCT05082259). ASTX660 is a small-molecule antagonist of all three clinically relevant IAPs (cIAP1, −2 and XIAP), which may offer advantages over first- and second-generation SMAC-mimetics, mainly by countering the redundancy of different IAPs in cancer cells. Pools of organoids (cell clusters) of the respective BP tumour organoids were injected orthotopically into the mammary fat pad of C57BL/6J females. When tumours reached 50–100 mm³, mice were either treated with vehicle or with ASTX660, αPD-1 or ASTX660/αPD-1 (Fig. 5A). Although ASTX660 and αPD-1 had negligible effects as monotherapies, ASTX660/αPD-1 achieved therapeutic responses in 45% of BP962-tumour bearing mice (Fig. 5B–D). Interestingly, the response to ASTX660/αPD-1 was dramatically improved under necroptotic conditions in the presence of emricasan, with a 71% response rate, curing 8 out of 17 animals and causing significant partial responses in 4 animals (Fig. 5E–G and Supplementary Fig. 6A). Animals cured by ASTX660/E/αPD-1 treatment were completely resistant to tumour re-challenge with BP962 tumour organoids (Fig. 5H–J), suggesting that these animals may have developed immune memory against tumour antigens. Immunologically protected animals also exhibited normal spleen weights, whereas BP962 tumour-bearing naïve mice displayed enlarged spleens (Fig. 5K). Importantly, other tumour lines with similar high RIPK1 expression and substantial macrophage infiltration to BP962 show comparable sensitivity to the combination treatment of

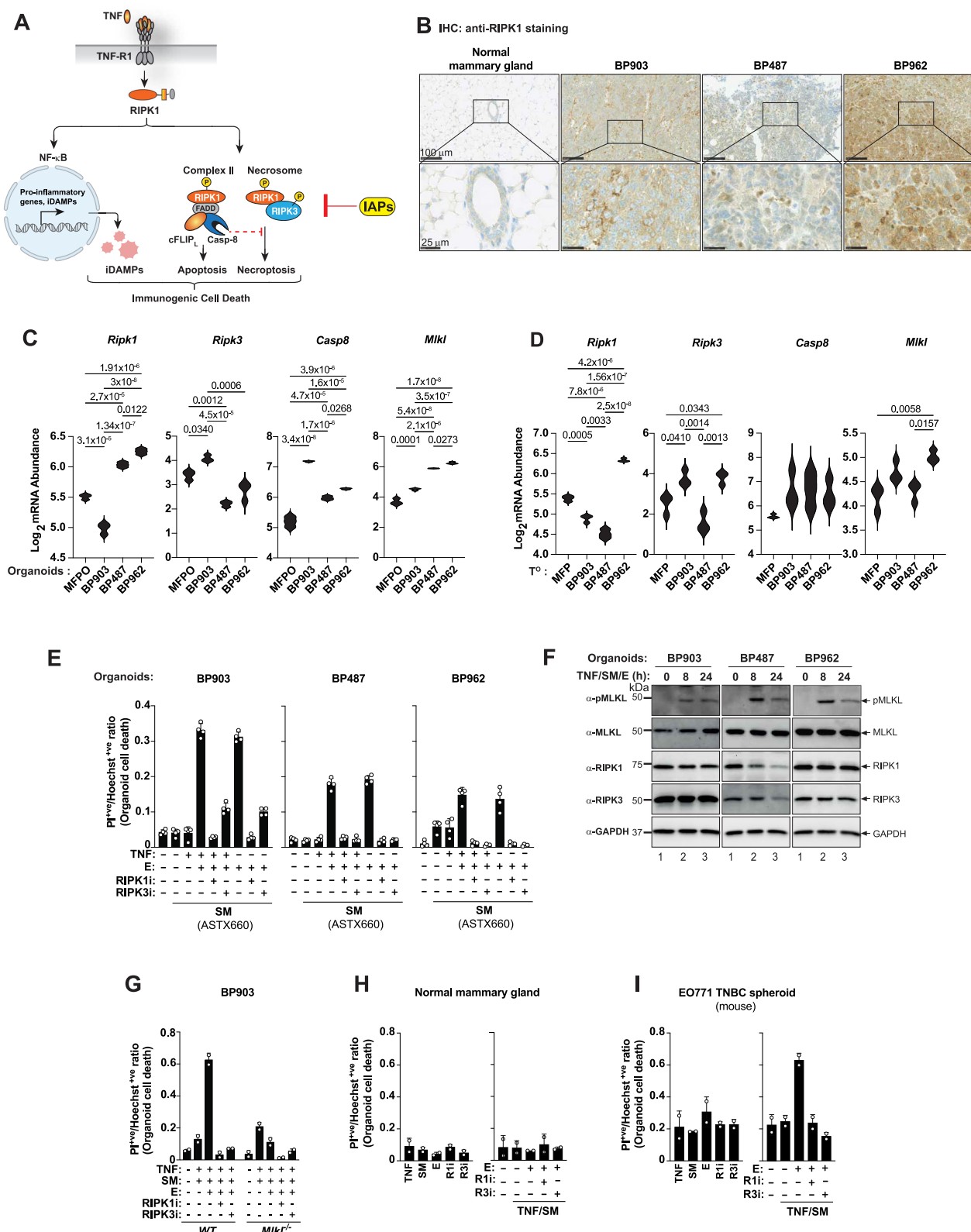

ASTX660/E/αPD-1 (Supplementary Fig. 6B–G), suggesting that elevated RIPK1 expression and pronounced macrophage infiltration may act as indicators of necroptotic responsiveness.

While treatment with ASTX660/αPD-1 caused significant responses in immune-infiltrated BP962 tumours, particularly under necroptotic conditions, the same treatments had minimal to no effect on the immunologically 'colder' T$^O$s (BP903 and BP487) (Supplementary

Fig. 7A–G). Co-treatment with an additional, second checkpoint inhibitor, anti-CTLA-4, provided no further benefit in BP487-tumour bearing animals (Supplementary Fig. 7H–K). Together, these data demonstrate that engaging necroptosis signalling significantly improves responses to immune checkpoint blockade-based treatments in immune-infiltrated/immune-escaped tumours (BP962), with high RIPK1 expression serving as a potential biomarker. However,

**Fig. 4 | RIPK1-mediated cell death in BP903, BP487 and BP962 organoids.**
**A** Schematic representation depicting TNF-mediated activation of NF-κB and cell death. **B** Representative IHC images showing RIPK1 expression (brown) in the indicated primary tumours (T$^P$s), representative of three independent tumours (one per animal). Scale bar: 100 μm for the top panel and 25 μm for the bottom panel. **C** Violin plots indicating RNA expression of *Ripk1*, *Ripk3*, *Casp8* and *Mlkl* in the indicated organoids (n = 3 independent samples per organoid). **D** Violin plots showing RNA expression of *Ripk1*, *Ripk3*, *Casp8* and *Mlkl* in the indicated T$^O$s (n = 3 independent tumours per T$^O$s). **E** Cell death assays of organoids treated for 24 h with the indicated agents. Each dot represents one technical repeat, with multiple three-dimensional fields per well quantified and averaged prior to plotting. The observed effects are representative of four independent biological experiments. **F** Western blot analysis of MLKL phosphorylation/activation in the indicated organoids, treated with TNF/SM/E for 0, 8 and 24 h. Samples derive from the same experiment but different gels for P-MLKL, MLKL and RIPK3 were processed in parallel. The P-MLKL membrane was subsequently re-probed for RIPK1 and then GAPDH, which served as the loading control. A representative result from three independent experiments is shown. **G** Cell death assays of the indicated organoids, treated for 24 h with the respective agents. Each dot represents one technical repeat as in (**E**). The observed effects are representative of two independent biological experiments. **H** Cell death assays of the indicated organoids, treated for 8 h as indicated. Each dot represents one technical repeat as in (**E**). The observed effects are representative of four independent biological experiments. **I** Cell death assays of E0771 cells treated for 4 h as indicated. Each dot represents one technical repeat as in (**E**). The observed effects are representative of two independent biological experiments. Data show mean ± SD. **C**, **D** p values were calculated using one-way ANOVA (Bonferroni multiple comparison test). Source data are provided as a Source Data file and in Zenodo: https://doi.org/10.5281/zenodo.18130193.

animals with less immune-infiltrated tumours showed only a small but statistically significant benefit from such treatments, even under necroptotic settings and double immune checkpoint blockade, highlighting the importance of stratifying patients based on their tumour infiltrating lymphocytes and potentially RIPK1 levels.

To assess treatment-related changes in CD8+ T cells of BP962 tumours, BP962 T$^O$s were exposed to a single treatment and harvested 9 days after treatment (Fig. 5L). Treatment with ASTX660 stimulated a marked infiltration of CD3+ and CD8+ T cells into the tumour under necroptotic conditions, whereas they were largely excluded in the controls (Fig. 5M, N and Supplementary Fig. 7L). The increase in CD8+ T cells in treated tumours correlated with a decrease in immunosuppressive CD163+ M2-like macrophages in the peritumoral area, where they were most abundant (Fig. 5O, P). In conclusion, treatment with ASTX660 under necroptotic conditions significantly enhanced CD8+ T cell infiltration into BP962 tumours, which was associated with a reduction in CD163+ M2-like macrophages. This suggests that ASTX660 not only promotes immune cell infiltration but also alters the tumour microenvironment, potentially improving the efficacy of immunotherapy by shifting the balance towards a more immune-activating environment.

**Boosting the immunogenicity of IAP antagonism through 'viral mimicry'**
To enhance the responsiveness of immunologically 'cold' tumours to IAP antagonism and ensure more potent anti-tumour immunity, we employed a strategy rooted in viral mimicry. Viral mimicry involves the activation of innate immune receptors to simulate an infection at the tumour site, thereby prompting an immune response. Such approaches include the provision of ligands for pattern recognition receptors (PRR). To this end we made use of the STING agonist diABZI[73]. STING agonists are emerging as powerful viral mimicry agents in cancer therapy, designed to activate the cGAS-STING pathway, thus triggering innate immune responses within tumours by simulating viral infection, thereby enhancing anti-tumour immunity[74–76] (Fig. 6A). First, we evaluated the ability of diABZI to drive STING signalling in BP903 and BP487 organoids in vitro. Treatment with diABZI readily triggered activation of IRF3 and interferon signalling (STAT1 activation) in both organoids (Fig. 6B). However, STING agonist alone showed little cell death activity in vitro. Importantly, when combined with IAP antagonism and emricasan, which induces cell death, addition of STING agonist did not further enhance cell death (Fig. 6C), which is consistent with its reported signalling axis[77]. Next, we evaluated whether diABZI treatment could enhance the anti-tumour efficacy of ASTX660 in tumours that are poorly infiltrated by immune cells, such as BP903 and BP487 T$^O$s. While treatment with diABZI alone had no effect on the growth rate of BP903 and BP487 tumours (Fig. 6D–G and Supplementary Fig. 8A–C), it significantly improved tumour control in combination with ASTX660, particularly under necroptotic conditions (Fig. 6D–J). This resulted in a survival benefit in both BP903 and BP487

T$^O$-bearing animals, with one complete tumour regression in each cohort that remained tumour-free beyond 60 days (Fig. 6G, J). Addition of αPD-1 did not generate a durable immune response when combined with diABZI (Supplementary Fig. 8D–L), suggesting that additional contextual factors, such as the spatial distribution of macrophages, may be required for achieving sustained tumour control.

To assess treatment-related changes of the immune landscape, we treated BP903 tumours with diABZI/ASTX660/E and analysed its effect 12 days later (Fig. 6K). Tumour analysis revealed that co-treatment reshaped the tumour immune micro-environment under necroptotic conditions, promoting a marked increase in effector lymphocytes, such as CD8+ T cells, γδ T cells and NK cells (Fig. 6L, M and Supplementary Fig. 9A–E) that exhibited key features of activation (CD44+) (Fig. 6N). Accordingly, we observed a marked rise in IFN-γ-producing NK cells and CD8$^+$ T cells within the tumour microenvironment following diABZI/ASTX660/E treatment in BP903 tumours (Fig. 6O). Importantly, we also observed an increased population of polyfunctional CD8$^+$ T cells, co-expressing IFN-γ and TNF, indicative of enhanced effector function (Fig. 6O), although the frequencies of Perforin- or Granzyme B-expressing cytotoxic cells remained stable (Supplementary Fig. 9F–I). Moreover, diABZI/ASTX660/E also caused a clear reduction in immunosuppressive PD-1+ NK cells and regulatory T cells (Tregs) (PD-1+ as well as CD69$^+$CD25$^+$) (Fig. 6P, Q). Overall, the data point to a more activated and functionally potent anti-tumour immune response induced by diABZI/ASTX660/E treatment. Taken together, treatment with diABZI/ASTX660/E significantly altered the tumour immune microenvironment in 'cold' tumours, promoting the infiltration and activation of effector lymphocytes and reducing the presence of immunosuppressive cells. This indicates a shift towards a more immunostimulatory and less immunosuppressive tumour environment, thereby favouring anti-tumour immunity.

**Necroptosis in tumour and stromal compartments is required to trigger effective anti-tumour immunity**
To determine whether the therapeutic benefit of immunogenic cell death relies on necroptotic signalling within tumour cells and/or the stromal compartment following ASTX660/E/αPD-1, we deleted MLKL in each context. We also investigated the role of RIPK1 in tumour cells. To this end we generated BP962$^{Ripk1-KO}$ and BP962$^{Mlkl-KO}$ organoids using CRISPR/Cas9 (Supplementary Fig. 10A–D). Loss of *Ripk1* profoundly impaired treatment efficacy with BP962$^{Ripk1-KO}$ tumours displaying a markedly blunted response to ASTX660/E/αPD-1, with no long-term survivors (Fig. 7A–C). In contrast, RIPK1-proficient BP962 tumours achieved durable tumour control under the same condition (see Fig. 5E–G), with 8/17 mice surviving long-term and showing evidence of immune memory (see Fig. 5H–J).

Next, we evaluated the role of MLKL-mediated necroptosis in tumour cells. Successful MLKL deletion and loss of necroptotic competence in BP962$^{Mlkl-KO}$ organoids were confirmed by Western blot and functional assays (Supplementary Fig. 10A, B). BP962$^{Mlkl-KO}$ tumours

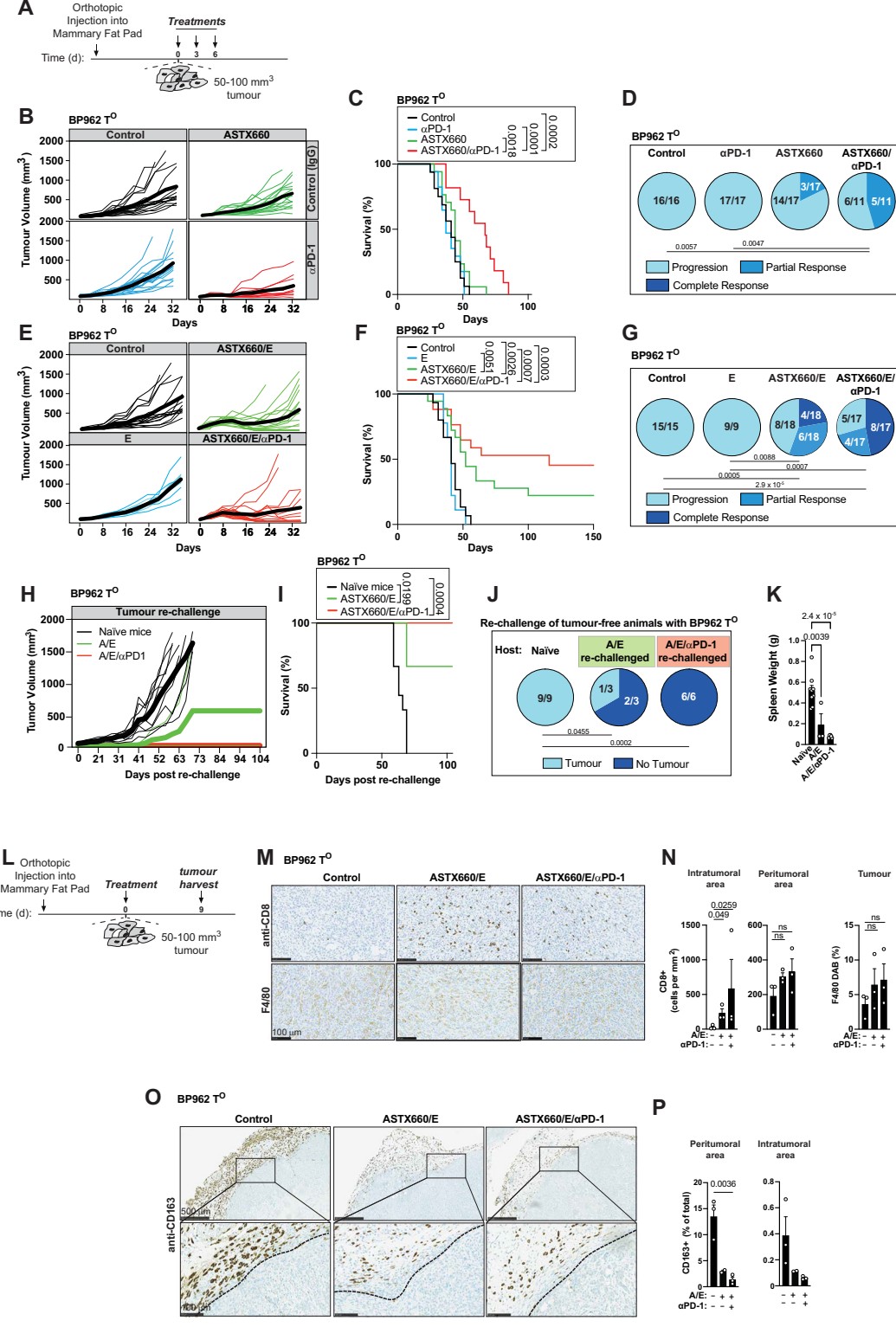

showed slightly slower growth in untreated mice, which was not apparent in BP962$^{Ripk1-KO}$ tumours (Supplementary Fig. 1OC, D). Intriguingly, upon therapy, BP962$^{Mlkl-KO}$ tumours remained responsive to ASTX660/E, however, unlike necroptosis-proficient tumours, co-administration of αPD-1 did not improve survival beyond ASTX660/E alone (Fig. 7E–G). This indicates that tumour-cell MLKL activity is required for αPD-1-mediated immunological potentiation. To determine

whether stromal necroptosis also contributes to therapeutic efficacy, BP962 organoids were transplanted into $Mlkl^{-/-}$ recipient mice. While ASTX660/E/αPD-1 still produced a measurable effect, its efficacy was substantially reduced in $Mlkl^{-/-}$ hosts, with no long-term survivors (Fig. 7H, I), compared to robust responses in WT mice (see Fig. 5E–G).

Together, these findings demonstrate that MLKL-mediated necroptosis in stromal cells is critical for durable anti-tumour

**Fig. 5 | Boosting immunotherapy by targeting IAPs. A** Schematic regimens of BP962 tumour-bearing mice. **B** Tumour growth: Control ($n = 16$), αPD-1 ($n = 17$), ASTX660 ($n = 17$) and ASTX660/αPD-1 ($n = 11$). Each thin line represents one mouse; thick lines indicate average tumour volume. **C** Kaplan−Meier survival for (**B**). **D** Proportion of mice with progressive disease, partial response, or complete response for each treatment group in (**B**). **E** Tumour growth of BP962 T⁰s, treated as in (**A**): Control ($n = 15$), αPD-1 ($n = 9$), ASTX660/E ($n = 18$) and ASTX660/E/αPD-1 ($n = 17$). Curves as in (**B**). **F** Kaplan−Meier survival for (**E**). **G** Response distribution for (**E**). **H** Tumour-rechallenge: naïve mice ($n = 9$), ASTX660/E-treated ($n = 3$) or ASTX660/E/αPD-1-treated ($n = 6$) tumour-free mice from (**E–G**). Tumour growth shown; thick lines denote average tumour growth. Mice were deemed tumour-free if no tumours developed for ≥2 months post-treatment. **I** Kaplan−Meier survival for (**H**). **J** Percentage of mice developing tumours versus remaining tumour-free after re-challenge. **K** Spleen weights after re-challenge in naïve vs. previously tumour-free mice; each dot represents one mouse. Data are mean ± SEM. **L** Treatment schedule for tumours analysed by immunohistochemistry (IHC). **M** Representative immunohistochemistry (IHC) staining of the indicated cell populations in treated tumours (brown, positive staining). Images are representative of three independent tumours derived from separate animals. Scale bar, 100 μm. **N** Quantification of CD8⁺ cells in peri- and intratumoural regions and F4/80⁺ cells in the intratumoural region, corresponding to (**M**). Each dot represents an individual tumour ($n = 3$ per group; tumours derived from independent animals). **O** Representative IHC images of CD163⁺ macrophages at the tumour periphery, representative of three independent tumours derived from separate animals. Scale bars, 500 μm (top) and 100 μm (bottom). **P** Quantification of (**O**). Each dot represents one tumour derived from an individual animal. Control ($n = 3$ tumours), A/E ($n = 2$ tumours), A/E/αPD-1 ($n = 3$ tumours). Data are mean ± SD. Statistics: log-rank (Mantel−Cox) (**C, F, I**), Fisher's exact (two-sided) (**D, G, J**), one-way ANOVA with Tukey's multiple comparisons (**K** and **N**; **N** analysed on log10-transformed CD8⁺ data), one-way ANOVA with Dunnett's multiple comparisons test (**P**). A ASTX660; E emricasan. Source data are provided as a Source Data file and in Zenodo: https://doi.org/10.5281/zenodo.18130193.

immunity, and that tumour-cell necroptosis is additionally required for full therapeutic benefit. Thus, necroptosis in both tumour and stromal compartments acts cooperatively to enable a maximally effective immune response to ASTX660/E/αPD-1.

## Discussion

The considerable heterogeneity observed in breast cancer subtypes represents a substantial challenge for treatment[78], as it is difficult to target the full spectrum of variability with conventional therapies. Immunotherapy has the potential to effectively address the diversity of tumour heterogeneity by targeting the immune system to recognise and combat a broad range of cancer cell variations. Although the use of immune activation represents a promising strategy to address this complexity, single genetically engineered mouse models (GEMMs) and clonal murine tumour models are inadequate for capturing this diversity, necessitating the utilisation of multiple genetic models to accurately reflect it and develop treatment protocols that more effectively engage a patient's immune system. Here, we demonstrate that the *Blg-Cre;Brca1^f/f^;p53^+/−^* tumour model produces a spectrum of genetically and phenotypically different invasive basal-like breast carcinomas. Although the same tumour suppressors are lost in the luminal progenitor cells of the mammary gland in every animal, tumours from different animals greatly vary in nuclear morphology and tissue microarchitecture, including differences in tumour/stroma ratio, immune composition and treatment responses−thus reflecting the heterogeneity of human basal-like breast cancers. However, the long and highly variable latency period for spontaneous tumour development and the inter-tumour heterogeneity of *Brca1^−/−^p53^−/−^*-tumour bearing animals presents a major challenge for conducting preclinical therapeutic experiments.

To address this issue, we developed a transplantation-based model whereby we derived cancer organoid lines from individual primary *Brca1^−/−^p53^−/−^* tumours. These organoid lines can subsequently be orthotopically transplanted into naïve immunocompetent female mice for the generation of pre-clinical cancer models that closely resemble human disease histologically, transcriptomically, and immunologically. Although individual lines significantly differ from one another, each line remains histologically stable over longer-term passage, and upon orthotopic transplantation into recipient females, generates a reproducible tumour within a 30−40 day period. Moreover, individual organoid-derived tumours are mosaic, in that engrafted cancer organoid cells are surrounded by genetically normal stromal cells, including immune cells. Most importantly, organoid-derived tumours bear a striking resemblance to its parental tumour from which the organoid line was derived. The organoid-derived tumours not only reproducibly replicate key histopathological and immunological features of their original primary tumours, but also those seen in human TNBC. For example, the microarchitecture of BP487 tumours closely mirrored clinical samples that present with interconnected and irregular clusters of malignant cells, a high collagen density in the stroma, and correspondingly low levels of tumour infiltrating lymphocytes, as well as a mixture of plump and spindly cancer-associated fibroblasts (CAFs). BP962 T⁰s have high levels of tumour-associated macrophages (TAMs), a situation that is frequently observed in TNBC patients where TAMs often represent 50% of all cells within the tumour microenvironment (TME)[79]. Intriguingly, we noticed heterogeneous expression of RIPK1 in these tumours, with a correlation between macrophage infiltration and elevated RIPK1 expression. We speculate that macrophage-produced cytokines, such as TNF, may stimulate compensatory upregulation of RIPK1 to protect tumour cells from cell death. While elevated RIPK1 levels may provide resistance to death ligands under basal conditions, they may also 'prime' these tumours for necroptotic activation upon appropriate triggers (ASTX660/E). Additionally, BP962 tumours harbour epithelial and mesenchymal cancer cell populations, a feature that is often associated with the cellular programme of epithelial-mesenchymal transition (EMT) and confers drug resistance, metastatic capacity and poor prognosis in TNBC[44,45]. Therefore, this preclinical transplantation-based platform provides an opportunity to create large murine patient cohorts with synchronous disease onset in which to evaluate treatment approaches in immunocompetent hosts. In this context, our work is complementary to previous efforts to build genetically defined organoid models[80,81] and living biobank repositories[82], as well as direct injection of organoids into the colonic submucosa[83,84]. Our study differs, however, in that we are using one and the same genetically engineered mouse model (GEMM) to generate a spectrum of variable TMEs in immunocompetent hosts.

The synchronous and highly reproducible characteristics of this transplantation-based TNBC model, in immunocompetent host animals, provide a rapid and cost-effective platform for the validation and optimisation of treatment strategies for TNBCs, particularly those designed to not only eliminate cancer cells but also activate the immune system against the tumour. Given that the transplantation-based BP organoid-derived tumours can accurately model human disease following orthotopic engraftment, we tested whether it is possible to evaluate the effects of immunogenic cell death to drive anti-tumour immune responses, in the presence and absence of ICB. To this end, we investigated the impact of RIPK1-induced immunogenic cell death. RIPK1 is a major stress sentinel, whose activation can drive the production of danger signals[14] as well as cell death in the form of apoptosis, necroptosis or pyroptosis[22,24,67,68,85–89]. IAPs are crucial gatekeepers of RIPK1's cytotoxic potential and are frequently overexpressed in breast cancer[26–31]. Simultaneously, IAP depletion in immune cells triggers activation of the non-canonical NF-κB pathway[90,91], generating co-stimulatory "danger" signals that facilitate dendritic cell (DC) maturation[92] and efficient cross-priming of effector

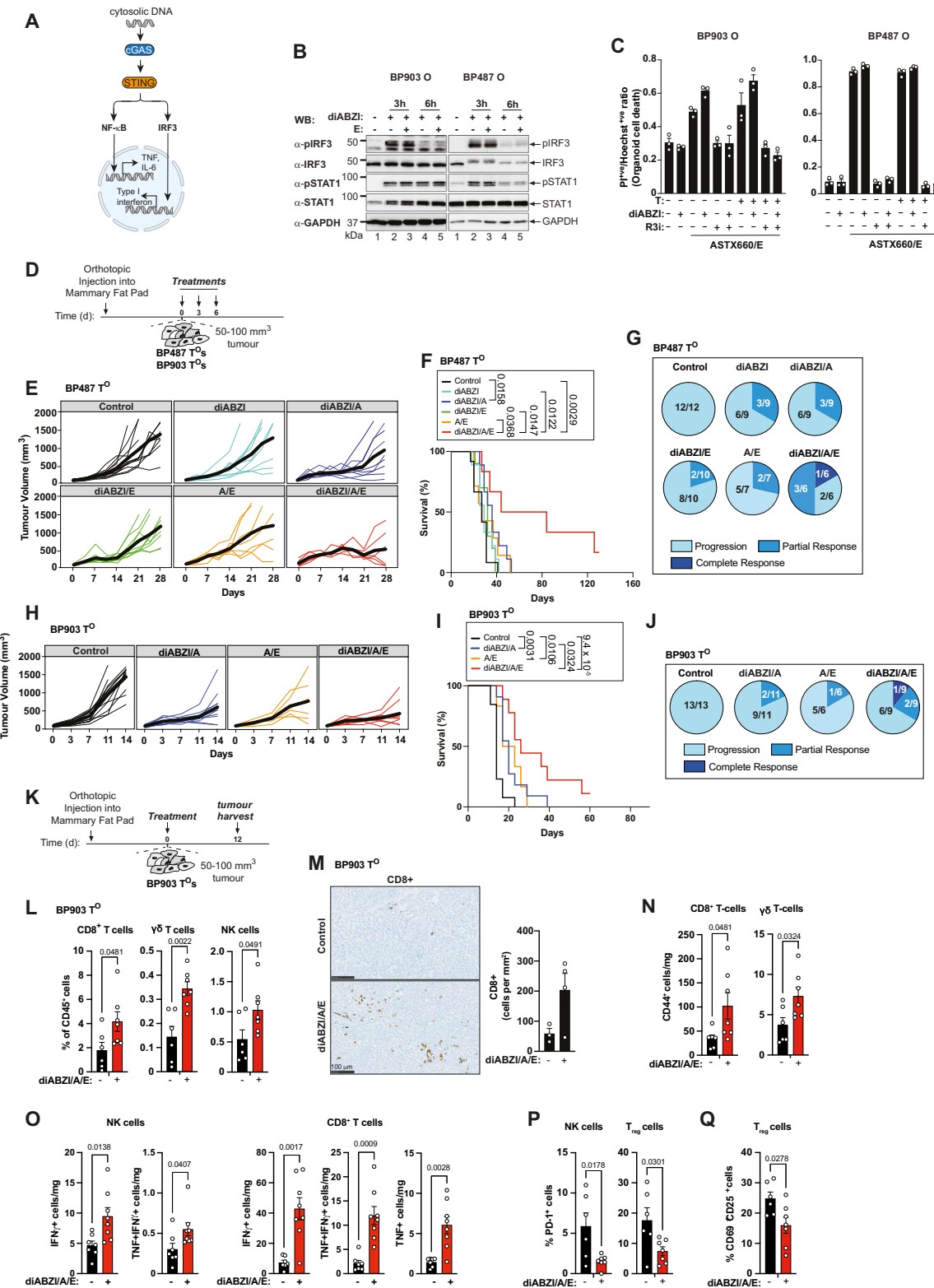

T cells[93]. This dual impact suggests that IAP antagonists have the potential to not only induce RIPK1-mediated immunogenic cell death, but also to amplify T cell responses. Therefore, combining IAP antagonists with immune checkpoint inhibitors (ICI) could effectively harness anti-tumour immunity, particularly in patients with treatment-resistant triple-negative disease, an approach that is currently being explored in the ASTEROID phase I clinical trial.

In co-clinical studies using mice to replicate the treatment protocol from the ASTEROID trial, we observed that the combination of IAP antagonism with αPD-1 significantly enhanced treatment responses, especially under necroptotic conditions, with a significant proportion of mice achieving a complete cure and the development of immune memory. However, only tumour-bearing mice with substantial immune cell infiltration within the tumour benefited from the

**Fig. 6 | Boosting the immunogenicity of IAP antagonism through 'viral mimicry'. A** Schematic of the cGAS/STING signalling pathway. **B** Western blot analysis of STING pathway activation in the indicated organoids, treated with the indicated agents. Samples derive from the same experiment and were run on separate membranes processed in parallel. For BP903, pSTAT1 was re-probed for IRF3. For BP487, pIRF3 was re-probed for pSTAT1. A representative result from three independent experiments is shown. **C** Organoid cell death after 24 h treatment as indicated. Each dot represents one technical repeat. The observed effects are representative of two independent biological experiments. **D** Schematic of the treatment regimen of BP487- and BP903-tumour-bearing C57BL/6J mice. **E** Tumour growth curves of BP487 T$^O$s, treated as indicated. Control ($n = 12$), diABZI ($n = 9$), A/E ($n = 7$), diABZI/A ($n = 9$), diABZI/E ($n = 10$) or diABZI/A/E ($n = 6$). Each line represents one mouse; thick lines indicate mean tumour volume. **F** Kaplan–Meier survival for (**E**). **G** Response distribution (progression, partial, complete response) for groups in (**E**). **H** Tumour growth curves of BP903 T$^O$s, treated as in (**D**): Control ($n = 13$), A/E ($n = 6$), diABZI/A ($n = 11$), diABZI/A/E ($n = 9$). Curves as in (**E**).

**I** Kaplan–Meier survival of BP903-tumour-bearing animals, treated as in (**D**). **J** Response distribution (progression, partial, complete response) for groups in (**H**). **K** Schematic of the treatment regimen for immune profiling experiments in BP903-tumour-bearing mice. **L** FACS analysis of BP903 T$^O$s showing percentages of the indicated cell populations. Tumours were treated as in (**K**) and harvested 12 days post-treatment. Each dot represents a tumour. Control ($n = 6$) and diABZI/ASTX660/E ($n = 7$). See Supplementary Fig. 9C for gating strategy. **M** Representative IHC images of CD8$^+$ cells (brown) in BP903 tumours. Control tumours ($n = 3$) and diABZI/A/E-treated tumours ($n = 4$) were analysed. Scale bar: 100 μm. Each dot represents the number of CD8$^+$ cells per tumour, with 3 sections analysed per tumour. **N–Q** FACS analysis of the indicated cell populations. Each dot represents a tumour. See Supplementary Fig. 9D for gating strategy. Data are mean ± SEM. *p* values were calculated using Log-rank (Mantel–Cox) test (**F**, **I**), unpaired *t*-test, two-tailed (**L–N**, **P**, **Q**) and Welch's *t*-test, two-tailed (**O**). A = ASTX660; E = emricasan; diABZI = STING agonist. Source data are provided as a Source Data file and in Zenodo: https://doi.org/10.5281/zenodo.18130193.

treatment. Mice with minimal tumour immune infiltration exhibited no response, even under necroptotic conditions and despite double ICB. This highlights the importance of stratifying patient selection for immunomodulatory treatments, based on the immune microenvironment of their tumour. Consistent with this, the E0771 model closely mirrors the BP962 phenotype, displaying a similar macrophage-rich microenvironment and responding even more robustly to combined ASTX660/E/αPD-1 therapy, likely reflecting its greater susceptibility to necroptosis.

A key question relates to the cellular origin of TNF that fuels the anti-tumour response upon IAP inhibition. In vitro, BP962 tumour organoids undergo cell death when treated with ASTX660 and emricasan, even in the absence of immune cells, suggesting that tumour cells themselves can produce TNF in an autocrine manner. This is consistent with previous reports demonstrating that IAP antagonists can trigger tumour-intrinsic TNF production[94,95], creating a self-amplifying death loop through RIPK1 activation. However, in vivo, immune-derived TNF is likely to play an additional role. CD8$^+$ T cells and myeloid cells are well-established sources of TNF in the tumour microenvironment, particularly after checkpoint blockade. Given that αPD-1 enhances T-cell activation, the therapeutic effect of ASTX660/αPD-1 is likely reinforced by TNF derived from infiltrating immune cells. The observation that αPD-1 improves efficacy only in necroptosis-competent tumours further supports the notion that immune-derived TNF amplifies inflammatory cell death and cross-priming in vivo. Although BP903 and BP487 are able to undergo necroptosis in vitro, they fail to respond to treatment in vivo (Supplementary Fig. 7A–G). This suggests that tumour cell necroptosis on its own does not determine responsiveness to αPD-1, and that other tumour- or microenvironment-related features are necessary. Accordingly, stromal necroptosis contributes more prominently to the anti-tumour response than tumour-cell necroptosis alone, although both are required. As CD8$^+$ T cells and macrophages are the principal source of TNF within the tumour, their abundance and spatial distribution likely determine the extent of necroptosis activation in stromal and tumour cells and, consequently, responsiveness to ICI. Together, these findings suggest a dual-source model in which tumour-intrinsic TNF may initiate RIPK1-dependent immunogenic cell death, while immune-cell-derived TNF further amplifies this response within the tumour microenvironment. Future studies using cell-type-specific TNF perturbation will be required to dissect the relative contribution of tumour- versus immune-derived TNF to treatment efficacy.

Supporting the notion that necroptosis preferentially benefits immunologically 'hot' tumours, we found that ASTX660/E treatment led to a significant increase in CD8+ T cells, coupled with a decrease in CD163+ M2-like macrophages. This finding implies that IAP antagonism not only enhances immune cell infiltration but also

remodels the tumour microenvironment, potentially improving the efficacy of immunotherapy by shifting the balance towards a more immune-stimulatory setting. It seems likely that the presence of tumour-infiltrating lymphocytes is crucial for the creation of a TNF- and interferon-rich environment, which in turn sensitises cancer cells, and stromal cells, to TNF-dependent and RIPK1-mediated cytotoxicity. This process facilitates the cytotoxic T lymphocyte (CTL)-mediated bystander killing of both cancer cells and potentially cancer-supportive stromal cells[96]. While interferon upregulates the necroptosis effector MLKL[97], TNF acts as the ligand for RIPK1 activation[67,68], driving RIPK1-mediated cell death. Consistent with this viewpoint, loss of sensitivity to TNF/IFN cytotoxicity perpetuates immune evasion and resistance to immunotherapy[7,54,98,99]. Lowering the TNF cytotoxicity threshold, such as through IAP antagonism, appears critical for effective anti-tumour immune responses and tumour eradication. Furthermore, additional mechanisms impact the cytotoxic capacity of TNF/IFN. Notably, cFLIP$_L$-driven caspase-8 activation can inhibit cell death by cleaving and inactivating RIPK1 and other death effectors like RIPK3 and CYLD[71,100–103]. Therefore, co-treatment with the clinical caspase inhibitor emricasan markedly enhances cancer cell sensitivity to RIPK1-mediated necroptosis, a lytic form of cell death that is markedly more immunogenic than apoptosis[14,16,57]. It is thought that this immunogenicity arises from the capacity of death receptors to activate NF-κB signalling while the cell is dying by necroptosis[16,57,104–107]. Both interferon signalling and NF-κB activation induce the production of iDAMPs, which, in conjunction with constitutive DAMPs, serve to robustly alert the immune system to danger[108]. This ensures that antigen-presenting dendritic cells simultaneously take up dead cells, sample cancer epitopes, and become activated to promote CD8+ T-cell cross-priming[21]. Consequently, therapies that induce necroptosis not only directly kill tumour cells but also reactivate the patient's immune system against the cancer[57]. While patients with immune-infiltrated tumours can benefit from approaches that lower the TNF/IFN cytotoxicity threshold, those with immunologically 'cold' tumours will require a different treatment strategy as they will not harbour enough CTLs to produce a sufficient amount of TNF/IFN for anti-tumour activity.

To 'heat up' tumours with less immune cell infiltration, we employed the principle of 'viral mimicry'. By exposing tumours to STING agonists, we were able to trigger the recruitment of effector lymphocytes, such as CD8+ T cells, γδ T cells and NK cells that exhibited key features of activation and anti-tumour function. As for immunologically infiltrated tumours, this required the lowering of the TNF/IFN cytotoxicity threshold via IAP antagonism. Moreover, promoting RIPK1-mediated necroptosis through caspase inhibition significantly enhanced tumour cell death and anti-tumour immunity.

Our findings highlight the importance of combining strategies that both lower the tumour's TNF cytotoxicity threshold and

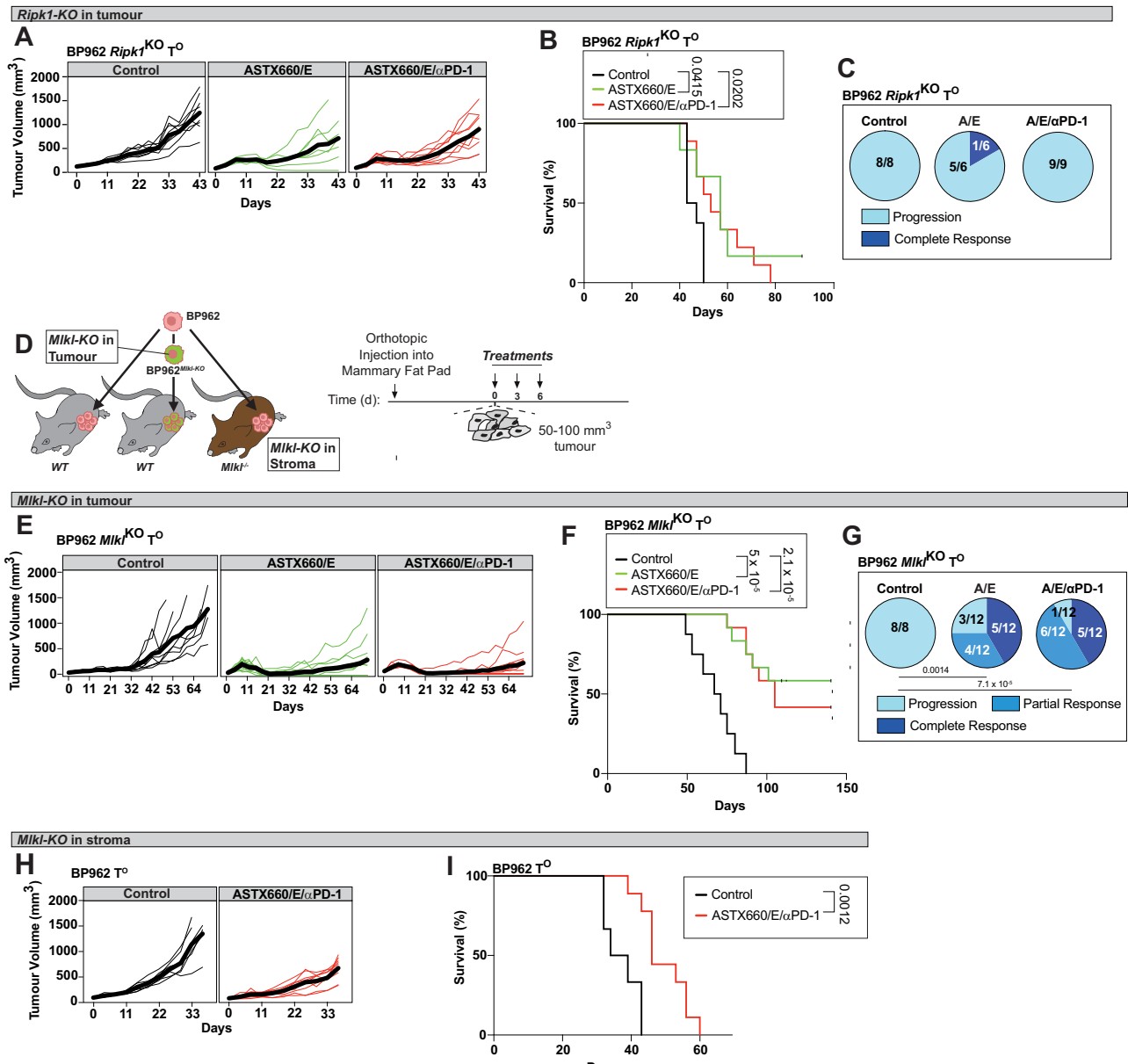

**Fig. 7 | Cancer cell intrinsic and tumour micro-environmental necroptosis contributes to the anti-tumour effect. A** Tumour growth curves of BP962$^{Ripkl\text{-}KO}$ in C57BL/6J mice: Control ($n = 8$), A/E ($n = 6$) and A/E/αPD-1 ($n = 9$). Each line represents one animal and thick lines represent average tumour growth. **B** Kaplan–Meier survival curves of BP962$^{Ripkl\text{-}KO}$ T$^O$ tumour-bearing animals. **C** Pie charts depicting the proportion of mice, which progressed, or fully responded to the indicated treatments. **D** Schematic representation depicting the experimental setup to evaluate the contribution of tumour versus stromal necroptosis. The treatment regimen is depicted on the right. **E** Tumour growth curves of BP962$^{Mlkl\text{-}KO}$ T$^O$s: Control ($n = 8$), A/E ($n = 12$) and A/E/αPD-1 ($n = 12$). Each line represents one animal and thick lines represent average tumour growth. **F** Kaplan–Meier survival curves of

BP962$^{Mlkl\text{-}KO}$ tumour-bearing animals, treated as in (**D**). **G** Pie charts depicting the proportion of mice from (**F**), which progressed, partially responded, or fully responded to the indicated treatments. **H** Tumour growth curves of BP962 T$^O$s grown in *Mlkl*$^{-/-}$ animals (see Fig. 5E–G for WT control). Treatments: Control ($n = 6$) and A/E/αPD-1 ($n = 9$). Each line represents one animal and thick lines represent average tumour growth. **I** Kaplan–Meier survival curves of BP962 tumour-bearing *Mlkl-KO* animals, treated as in (**D**). A = ASTX660; E = emricasan *p* values were calculated using Log-rank (Mantel–Cox) test (**B**, **F**, **I**) and Fisher's exact test, two-sided (**G**). Source data are provided as a Source Data file and in Zenodo: https://doi.org/10.5281/zenodo.18130193.

enhance the recruitment of tumour-infiltrating lymphocytes. Achieving both objectives is essential to improving the efficacy of ICIs like PD-1/PD-L1 and CTLA-4 blockers, in most tumours. Increasing the density and functionality of TILs alone, while essential, will not be sufficient to drive consistent anti-tumour immunity. In addition, lowering the threshold of RIPK1 activation, and fuelling its ability to drive necroptosis will significantly boost anti-tumour immunity. Collectively, the transplantation-based BP model will help to identify combination treatments that are more

effective in mobilising a patient's own immune system to eradicate tumours.

## Methods

All research reported in this study was approved by the Animal Welfare Ethical Review Body (AWERB), and conducted within the guidelines of UK Home Office in accordance with Animals (Scientific Procedure) Act (ASPA) 1986, amended 2012 and the institutional guidelines of the Institute of Cancer Research. Further, all animal experiments were

conducted in accordance with the Animal Research: Reporting of In vivo Experiments (ARRIVE) guidelines to ensure reproducibility and transparency[109].

## Reagents, chemicals and antibodies
See Supplementary Data 1.

## Generation of organoid lines
Organoid lines were generated using an adaptation of a previously described method (Duarte et al.[110]). Organoids were derived from tumours that had spontaneously developed in *Blg-Cre;Brca1^{f/f}pS3^{+/-}* mice. Tumours were excised once they reached a maximum diameter of 8–12 mm and were then mechanically disaggregated using scalpel blades. Subsequently, they were digested in RPMI medium containing enzymes from the Tumour Dissociation Kit for 41 min at 37 °C, with the GentleMACS Octodissociator (Miltenyi) set to varying rotation speeds. The digested suspension was filtered through a 70 μm cell strainer (Falcon), then centrifuged at 1400 rpm for 10 min. The pellet was washed twice in basal ADDF+++ media (Advanced DMEM/F12 supplemented with 10 mM HEPES, 1% GlutaMAX, and 1% penicillin/streptomycin), then resuspended in mouse mammary gland organoid medium ADDF+++ media, supplemented with 2% B27 supplement (Gibco), 125 μM N-acetyl-L-cysteine (Sigma), and 50 ng/mL mEGF (Thermo Fisher Scientific), mixed 1:1 with Growth Factor reduced Matrigel (SLS). The organoid:matrigel mix was seeded into 40 μl domes per well into a 24-well suspension plates (Greiner Bio-One). After incubation at 37 °C for 30 min, 500–750 μl of ADDF+++ media containing 10 μM ROCKi (Medkoo) was added per well. Healthy mammary tissue organoids were derived using an adaption of the previously described method[110] and the dissociation of mouse mammary tissue using EpiCult^{TM}-B. Specifically, mammary fat pads were isolated from C57BL/6J mice and minced to a paste using scalpels before digesting in EpiCult^{TM}-B digestion medium containing 1x Collagenase/hyaluronidase (Stemcell Technologies, #07919), 5% FBS and 1% penicillin/streptomycin during a 41 min incubation at 37 °C, with varying rates of rotation in the GentleMACS Octodissociator (Miltenyi). After digestion, the suspension was mechanically disrupted with a 1 ml pipette tip and pelleted at $350 \times g$ for 5 min and filtered through a 70 μm cell strainer (Falcon). The suspension was centrifuged at 1400 rpm for 10 min and washed twice in basal ADDF media before resuspension in Normal Mammary media of which the composition was based as described before[111] with Neuregulin substituted with mEGF (ADDF+++ media; supplemented with 2% B27, 1% N2 (Thermo Fisher Scientific), 10 ng/ml mEGF, 10 ng/ml FGF basic (Peprotech), 4 μg/ml Heparin (Sigma) mixed 1:1 with Growth Factor reduced Matrigel (Corning, #356237). The organoid:matrigel mix were seeded into 40 μl domes per well into a 24-well suspension plates (Greiner Bio-One). After incubation at 37 °C for 30 min, 500–750 μl of Normal Mammary media containing 10 μM ROCKi was added per well.

## Maintenance of organoid lines
Once established, organoids were cultured in a humidified incubator with 5% $CO_2$ at 37 °C. Organoids were passaged as required at a ratio of 1:2–1:10, depending on the line. To passage, organoids were extracted from Matrigel and dissociated to single cells in TrypLE for 10 min at 37 °C. Single cells were centrifuged at 1400 rpm for 4 min and the cell pellet was resuspended in either a 50% or 75% Matrigel:ADDF+++ mix and then plated as described above. Organoids were frozen 72 h post passage in FBS supplemented with 10% DMSO. For all experiments, organoids were used between passages 5 and 25. The E0771 cell line was cultured in RPMI1640 (Gibco), supplemented with 1% penicillin/streptomycin and 10% FBS. All organoids were screened for pathogens by an external provider (Charles River, UK), using the 'Charles River Mouse/Rat Comprehensive CLEAR Panel w/ C. bovis'. All cells and organoids were regularly screened for mycoplasma using the MycoAlert Mycoplasma Detection Kit (Lonza).

## Generation of CRISPR knockout organoids
Guide RNA preparation was described before[60]. For MLKL knockout, organoids were broken down into single cells using TrypLE and $1 \times 10^6$ cells were seeded per well into Matrigel coated 6 well plates. After 24 h, cells were transiently transfected with Lipofectamine 3000 (Invitrogen). After 48 h, GFP positive cells were FACS sorted, seeded back into Matrigel and screened for gene knockout. For RIPK1 knockout, annealing of relevant guides and ATTO-550 tracer (IDT) was performed through annealing in a 95 °C−RT ramp programme, over the course of 30', according to manufacturer's instructions. RNP complex formation was performed by supplementing Cas9 at a final concentration of 63 μM, for 20' at RT, before electroporation. Single suspensions of $2 \times 10^5$ cells were resuspended in 27 μL of Neon Resuspension R buffer (Invitrogen), 3 μL of RNP complex and 6 μL of Nuclease-Free Duplex Buffer (IDT) and then electroporated using 10 μL Neon NxT tips (Invitrogen) under a programme of 1 pulse at 1700 V and 20 ms pulse width programme. Immediately after electroporation cells were seeded in P/S-free ADDF medium, spun down, and replated in 3D O/N. The following day, cells were freed from Matrigel through TrypLE digestion, FACS sorted for ATTO-550 positivity, and re-plated in 3D. Cells were cultured for 1 week in presence of P/S-free ADDF medium, before validation by Western Blot.

## PCR genotyping
Organoids were homogenised using a Qiashredder (Qiagen). For ex vivo mouse tissues, samples were snap frozen immediately after dissection, then a 20–30 mg fragment of tissue was disrupted and homogenised using a Precellys bead mill (Bertin). DNA was extracted from the homogenised samples using the DNeasy Blood and Tissue Kit (Qiagen), as per manufacturer's instructions. PCR reactions were performed according to the mouse genotyping protocols published by The Jackson Laboratory (Jackson Laboratories 2019), using the KAPA2G Fast HotStart PCR Kit (Kapa Biosystems), with primers (see Supplementary Data 1). DNA electrophoresis was run in 2.5% agarose gels (Invitrogen) and gels were imaged using a ChemiDoc Touch Imaging System (Bio-Rad).

## RT-qPCR
RT-qPCR was performed using two TaqMan assays (Thermo Fisher Scientific); one with the target sequence in exon 19 (present in all alleles) and one with the target sequence in exon 22 (which is present if the floxing event has not occurred, and absent if the floxing event has occurred)[43]. RT-qPCR was performed on RNA extracted from tail tips of wild-type C57BL/6J mice, organoids, and organoid-derived tumours. Prior to RNA extraction, mouse tissues (tumours and tail tips) were mechanically disrupted and homogenised using a Precellys 24 tissue homogeniser (Bertin). For organoids $1 \times 10^6$ cells were collected and homogenised using a Qiashredder (Qiagen). RNA was extracted from homogenised samples using the RNeasy Mini Kit (Qiagen) as per manufacturer's instructions. RNA quantity and quality were measured using a NanoDrop 8000 Spectrophotometer (Thermo Fisher Scientific). cDNA was synthesised from RNA using the High-Capacity cDNA Reverse Transcription Kit (Applied Biosystems). For *Brca1* exon 19 and exon 22, qPCR was performed using TaqMan Gene Expression Assays (Applied Biosystems) as previously described[43]. RT-qPCR results were analysed using the DDCt method. Ct values were calculated from triplicate repeats for each sample. The Ct value of a housekeeping gene (*βActin*) was subtracted from the Ct value of each sample, to calculate the DCt value. The DCt value for each sample was then normalised to a control sample by subtraction to calculate DDCt. The fold change in expression was calculated as the negative exponent of this value (2-DDCt). RNA was

extracted from homogenised BP903 tumour fragments using the Allprep DNA/RNA Micro kit (Qiagen) and converted to cDNA using QuantiTect Reverse Transcription Kit (Qiagen), according to manufacturer's instructions. RT-qPCR and data analysis were performed as previously described[112]. For BP903 T⁰s, mRNA was extracted 12 days post treatment with a single injection of vehicle control or diABZI/ASTX660/E. The relative mRNA expression of indicated genes was measured using Taqman probes (Thermo Fisher Scientific): *Actb* (Housekeeping gene (Mm00607939_s1), *Ifng* (Mm01168134_m1), *Perforin* (Mm00812512_m1), and *Ccl5* (Mm01302427_m1).

## Bulk RNA-seq

RNA was extracted using the Allprep DNA/RNA Micro kit (Qiagen) from snap frozen samples of the primary tumour, early passage established normal and tumour organoids (passage number 10-20), organoid-derived tumours, as well as normal murine mammary fat pads (excluding lymph nodes). Three independent samples were analysed per group, except for the primary tumour, for which a single sample was analysed. For organoids, $1 \times 10^6$ cells were collected and homogenised using a Qiagen Qiashredder. For mouse tissue, samples were snap frozen immediately after dissection, then a 20–30 mg fragment of tissue was disrupted and homogenised using a Precellys bead mill (Bertin). RNA quantity and quality were determined using the Nanodrop spectrophotometer (Thermo Fisher Scientific). Strand-specific mRNA libraries were prepared using the NEBNext Ultra II Directional RNA Library Prep Kit for Illumina (NEBNext Poly(A) mRNA Magnetic Isolation Module; New England Biolabs), from 500 ng of RNA per sample. The quantity and quality of the libraries were analysed using the 2100 Bioanalyzer system (Agilent). Sequencing was performed using the Illumina NovaSeq 6000, on an S1 100 bp PE Nova flowcell, to achieve 7.5 Gigabases per sample.

## Copy-number analysis

Copy-number aberrations were calculated using CNVkit (v0.9.9[113]) and Control-FREEC (v11.5[114]). For CNVKit, gains/amplifications and losses/deletions were defined as log2ratio > 0.5 and log2ratio < −0.3, respectively. For Control-FREEC, default gain/amplification and loss/deletion filters were used. Results from the two callers were converted to R GRanges objects and subsequently intersected keeping the common copy-number calls only. Preprocessing of whole exome sequencing dataset was performed as previously described[115] using the mouse reference genome (GRCm38).

## Computational analysis of RNA sequencing data

For raw sequence quality control, FastQC and FastQ Screen were run on all sample FASTQ files, and a summarised report was generated using MultiQC[116,117]. FASTQ reads were trimmed using Trim Galore (v0.6.6). Paired-end reads (100 bp long) were aligned to the mouse reference genome (version GRCm38), using STAR (v2.7.6a) with quantMode GeneCounts and twopassMode Basic alignment settings[118]. The annotation file used for feature quantifications was downloaded from GENCODE (v17) in GTF file format. Differential mRNA abundance analysis was performed using the edgeR package[119] in R (version 3.6.0). Results were annotated using ENSEMBL gene annotations from R package org.Mm.eg.db. Genes were considered statistically significant if the log2 fold change was greater than 1, and the false discovery rate-adjusted *p* value was below 0.05. Principal component analysis (PCA) was performed using custom libraries in R statistical environment (v.3.6.0). Normalised data was used for PCA using R function FactoMineR::PCA for plotting PC1 and PC2. Differences in the expression of stromal elements was interrogated using the stroma-derived prognostic predictor (SDPP) gene set[120]. Samples were classified into breast cancer subtypes based on their gene expression profiles using the AIMS gene classifier[121]. A mouse-specific mammary tumour classifier was created using centroids of a previously published microarray

dataset[122]. The MHC class I antigen presentation pathway was analysed by comparing the differential expression of various organoids, using a gene list from Nanostring's pan-cancer/immune gene sets (https://nanostring.com/products/ncounter-assays-panels/oncology/pancancer-immune-profiling/) (Supplementary Data 2). The immune checkpoint gene expression was analysed by comparing the differential expression of the indicated organoids.

## Gene set enrichment analyses (GSEA)

Macrophage-related gene sets were obtained from MSigDB[123] by querying mouse collections for the keyword "macrophage" and further filtering to include only those containing the term in their name. Gene Set Enrichment Analysis (GSEA) was performed in R 4.2.1 with ClusterProfiler using fold-change values from differential expression comparisons of different T⁰s (Supplementary Data 2).

## Cell type deconvolution with CIBERSORTx

To perform cell type deconvolution of bulk RNA samples, cell-type specific transcriptomes were obtained from the *Brca1⁻/⁻p53⁻/⁻* TNBC mouse model[124]. scRNA-seq counts were converted to transcripts per million (tpm) and the dataset was down sampled to 2500 cells to increase computing efficiency. A signature matrix was then created using a CIBERSORTx[125] singularity container with default settings. Subsequently, fractions mode was employed for cell type proportion deconvolution using relative-mode, 100 per mutations, rmbatchS-mode and disabling quantile normalisation.

## Lysate preparation and Western blot analysis

Organoids embedded in Matrigel were washed with PBS and frozen at −80 °C overnight. Once thawed at room temperature, the organoids and Matrigel domes were lysed using S-Buffer (20 mM Tris pH 8.0, 40 mM Na pyrophosphate, 50 mM NaF, 5 mM MgCl₂, 100 μM NaVanadate, 10 mM EGTA, 1% Triton-X-100, 0.5% NaDOC) containing protease inhibitor cocktail and phosphatase inhibitors. Organoids and Matrigel were mechanically disrupted by pipetting and spun down at $13{,}200 \times g$ at 4 °C for 10 min to coalesce the Matrigel. 6x sample buffer (350 mM Tris pH 6.8, 35% glycerol, 10% SDS, 0.6 M DTT, 180 μM bromophenol blue) was added to the supernatant and the samples were heated to 85 °C for 5 min and then resolved on NuPAGE Novex 4–12% Tris-Gly 1.0 mm gels in Tris-Glycine buffer (Thermo Fisher). After transfer onto polyvinylidene difluoride (PVDF), membranes were blocked with 5% BSA and then probed with antibodies as indicated.

## Cell death assay with organoids

Organoids were seeded as single cells into 384 well plates (Greiner, #781091), containing $5 \times 10^3$ cells embedded in 4–5 mg/ml Matrigel, ADDF+++ media mix per well and cultured for at least 72 h before drug treatments. Drug concentrations used: TNF (10 ng/ml), IAP antagonist (SM164, 1 μM, and ASTX660, 1 μM or 10 μM), Emricasan (5 μM), RIPK1 inhibitor (GSK'693, 100 nM), RIPK3 inhibitor (GSK'872, 10 μM) and STING agonist (diABZI, 10 μM). ASTX660 was used throughout, except for Fig. 4F, H, I and Supplementary Figs. 5D and 10B, where SM164 was used. At experimental endpoint, wells were supplemented with propidium iodide (PI) (1 μg/ml, Sigma-Aldrich), and Hoechst-33342 (H) (0.5 μg /ml) (Thermo Fisher Scientific) and incubated at 37 °C for 1 h. To image cell death (PI uptake) of organoids, we used the ImageXpress Confocal High-Content Imaging System (Molecular Devices) with a Nikon Plan Fluor Ph1 DLL 10x/0.3 NA air objective. The system is equipped with a Lumencor LED light source and an Oxford Nanoinstruments Zyla camera. Images of the Hoecsht (405 nm ex/461 nm em) and PI (560 nm ex/ 615 nm em) were captured at 400 ms and 300 ms exposures, respectively, with 2 or 4 fields of view imaged per well and a z-stack of 5 μm steps over ~150 μm range, which was subsequently compressed to 2D maximum intensity projections with each field of view typically capturing ~20–50 organoids, with each organoid

consisting of ~100 cells). A custom module workflow was built using the MetaXpress analysis software (Molecular Devices). The workflow consisted of 9 steps. Step 1: 'Setup' module defined the channels used for analysis, 'DAPI' and 'Texas Red'. Step 2: 'Simple Threshold' module to segment the organoid structures using the DAPI channel to create an object mask of the organoid total area. The threshold was adjusted as required depending on signal intensity. Step 3: 'Filter Mask' module determined the size of organoid structures (or objects) to be included for analysis, and enabled exclusion of debris or single cells and adjusted as required between experiments. Step 4: 'Grow Objects' module followed by Step 5: 'Fill Holes' module improved segmentation by merging nuclei and filling in any space created by the presence of a lumen, to create complete objects for the mask. For the Grow objects, we consistently increased by 3 pixels. Step 6: 'Remove Round Objects' module removed all partially segmented organoids obstructed by the edge of the imaging field of view to complete the first organoid mask. Step 7: 'Find Round Objects' module was used to segment PI positive nuclei using thresholding on channel 2 to create the second mask of organoid cell death. Step 8: 'Grow Objects' module expanded the segmented PI positive objects by 3 pixels. Step 9: 'Measure Mask' module measured the total area of PI stain overlayed within the defined segmented total viable objects from channel 1. The above analysis pipeline outputs a large table containing the total area measurement for both the first organoid mask (total organoid area) and second organoid cell death mask (PI area only) for every object measured, as well as other related data. To automate and streamline the analysis, a python script was written in Jupyter Notebook to clean up the data table, calculate the mean PI/DAPI ratio per field of view and export the rearranged table into a format suitable for downstream analysis. One well typically corresponds to two or more three-dimensional fields of view within Matrigel. For each field, the mean PI/DAPI staining ratio was calculated across all quantified organoids. Organoid viability was measured using CellTiter-GLO assay (Promega). Briefly, organoids were seeded as single cells into 384 well plates (Greiner, #781091), containing $5 \times 10^3$ cells embedded in 4–5 mg/ml Matrigel, ADDF+++ media mix per well and cultured for at least 72 h before treatment with titrated Tamoxifen and Fulvestrant for 7 days. Media was refreshed after 3 days with media containing freshly prepared drugs. At experimental endpoint, media was aspirated and 50 ul neat CTG reagent was added per well and the plate was rocked gently for 1 h. Luminescence was measured using Victor X plate reader (PerkinElmer) and Data was normalised to the DMSO control to calculate cell viability.

## Flow cytometry analysis

To determine immune infiltration, tumours were harvested and single cell suspensions were prepared. To this end, tumours were perfused with 1 ml/200 mg digest buffer (1 mg/ml Collagenase II (Merck), 0.1 mg/ml DNAse I (Merck), and 0.1% BSA in RPMI) before being minced into 1–2 mm pieces and incubated at 37 °C for 1 h shaking at 180 rpm. Digestion was inactivated with 20 mL MACS buffer (Miltenyi autoMACS running buffer Miltenyi, 130-091-221), and tissue suspension passed through a 70 μm filter. Cell suspensions were spun down at 350 rpm for 15 min at 4 °C, and supernatant aspirated. Cell pellets were resuspended in 500 ml red blood cell lysis buffer (1x BD PharmLyse 555899 in $H_2O$) and incubated at RT for 2 min. Lysis was inactivated with 2 ml HBSS and cells pelleted at 350 rpm for 6 min at 4 °C, before aspiration of supernatant. Pelleted cells were resuspended in 500 μl viability dye (1:2000 dilution eBioscience™ Fixable Viability Dye eFluor™ 506, Thermo Fisher 65-0866-14 in HBSS) and incubated in the dark at RT for 20 min. Cells were washed with 2 ml MACS buffer and pelleted at 350 rpm for 6 min at 4 °C, before aspiration of supernatant. Pelleted cells were resuspended in 50 μl Fc block (1:50 dilution in MACS buffer of Purified Rat Anti-mouse CD16/CD32, BD Biosciences 553142), and incubated in the dark at 4 °C for 20 min. Antibody cocktail was

prepared in a staining volume of 50 μl MACS buffer and added directly to cells in Fc block, prior to a further 30 min incubation. Stained cells were washed with 300 μl MACS buffer and pelleted as before. Cell pellets were fixed in 2% PFA in MACS buffer, in the dark at 4 °C for 20 min. Finally, cells were washed as before, and resuspended in MACS buffer for flow cytometry analysis. 123 count eBeads (Thermo Fisher 01-1234-42) were added to each sample to calculate cell counts. To quantify changes in the immune landscape following treatment, tumours were harvested in ice-cold PBS from mice 12 days after treatment. Tumours were mechanically dissociated with scissors and enzymatically digested (30 min, 37 °C), in PBS containing Trypsin-Versene (in house), 0.5 mg/ml Collagenase type I (Sigma, cat# C2674), 400 μg/ml Dispase type II (Sigma, cat# D4693), 1 mg/ml DNase type I (Roche, cat# 10104159001). Tumour suspensions were passed through a cell strainer (70 μm) into PBS (2% FBS & 2 mM EDTA), before centrifugation (1500 rpm, 10 min, 4 °C). Pellets were resuspended in PBS-FBS 2% + Fc block (1:100, CD16/CD32, BD Biosciences cat# 553142) for 10 min (4 °C), before surface staining for 30 min at 4 °C with the appropriate antibody cocktail containing the viability dye (1:1000 eBioscience™ Fixable Viability Dye eFluor™ 780, cat# 65-0865-18). Cells were either fixed in 2% PFA for 20 min or fixed and permeabilised following the manufacturer's protocol (eBioscience, cat# 00-5523-00) prior to staining with anti-Granzyme B, anti-Foxp3 and Ki67 antibodies. Flow cytometric analyses were carried out with a FACSymphony A5 (BD Biosciences) with FACSDiva software. 123 count eBeads (Thermo Fisher, cat# 01-1234-42) were added to each sample to calculate cell counts. Data were analysed with FCS Express 7 software. The full list of antibodies used in this study can be found in the supplementary resources table.

## Immunofluorescence

Immunofluorescence analysis was conducted as described previously[126]. Briefly, organoids were seeded as single cells into 96 well plates (Greiner, #G655090), containing $5 \times 10^3$ cells embedded in 4–5 mg/ml Matrigel, ADDF+++ media mix per well and cultured for 7 days (tumour organoids) or 12 days (Mammary fat pad organoids) before fixing with 4% PFA for 30 min. Organoids were blocked and permeabilised with Blocking Buffer (0.3% Triton X100, 1% BSA in PBS) for 1 hr. All primary and secondary antibodies were used at 1:100 dilution in Blocking Buffer. DAPI was used at 1:1000 in Blocking Buffer. Antibodies used can be found in Supplementary Data 1. Organoids were fixed and Immunofluorescent labelling of FFPE sections was conducted as previously described[127]. Confocal imaging was performed using a Leica SP8 confocal microscope. Images were acquired at a resolution of 2048 × 2048 pixels with an 8-bit depth and a scan speed of 400. Imaging was conducted using a 20 × 0.8NA air objective (zoom 2.16) for BP487, BP903, and WTO samples, and a 40 × 1.3NA oil objective (zoom 1.5) for BP962 samples. Z-stack images were collected for all conditions. Maximum intensity projections were generated from z-stacks using Fiji software for presentation as two-dimensional images.

## Opal 6-plex

Three μm FFPE tissue sections were baked for 1 h and run on the Leica Bond Rx platform. Several cycles of sequential staining were conducted by using Epitope Retrieval solution 1 for 20 min between each antibody, apart for the last antibody in the sequence where Epitope Retrieval solution 2 was used for 30 min. Antibodies were applied with Opal™ pairings in the following order: Panel 1—Lyve1 (ab33682) 1:1500 with Opal 520 1:300, F4/80 (ab300421) 1:1000 with Opal 690 1:150, CD31 (ab182981) 1:1000 with Opal 480 1:100, Ki67 (ab15580) 1:1000 with Opal 570 1:300, CD45 (70257S) 1:100 with Opal 620 1:300 and CD19 (ab245235) 1:200 with TSA-DIG 1:100 followed by anti-DIG-Opal780 1:25. Panel 2—Ly6G (87048S) 1:100 with Opal 520 1:300, F4/80 (ab300421) 1:1000 with Opal 690 1:150, αSMA (M0851) 1:250 with Opal

570 1:300, CD3 (ab134096) 1:500 with Opal 480 1:100, CD45 (70257S) 1:100 with Opal 620 1:300 and CD19 (ab245235) 1:200 with TSA-DIG 1:100 followed by anti-DIG-Opal780 1:25. Bond anti-rabbit Polymer was used as the secondary antibody for antibodies raised in rabbit. Horse anti-Mouse IgG (Rat adsorbed)-Biotinylated secondary antibody (BA-2001) 1:400 and Streptavidin-Peroxidase (P0397) 1:500 were used as the secondary antibodies for antibodies raised in mouse. Slides were scanned using the PhenoImager HT (formerly Vectra Polaris). Spectral unmixing and auto-fluorescence removal were performed using the Phenochart™ and Inform™ softwares.

## Histopathology, sample preparation and staining

Tumour material was fixed in 10% neutral buffered formalin for 24 h. Subsequently, tumours were washed in PBS followed by processing from 50% EtOH through to embedding in paraffin wax. Organoids were fixed in 10% neutral buffered formalin for 15 min, embedded in Histogel (Richard-Allan Scientific) then processed to wax embedding as above. For IHC staining, 4 µm sections were cut onto adhesive slides, air dried and baked at 60 °C to enhance adhesion and stored at 4 °C prior to staining. Prior to IHC staining, heat induced epitope retrieval (HIER) and deparaffinisation was performed using the Dako PT LINK module for 20 min at 97 °C with a 20 min warmup and then 20 min cooldown to 65 °C, in either pH 6 or pH 9 Dako Target Retrieval Solution (Agilent Technologies). Endogenous peroxidases were blocked using Agilent REAL peroxide block. All staining was performed on the Dako Link48 or Ventana Discovery Ultra automated immunostaining platform, which, for RIPK1, was similar to the one described in ref. 128. For staining carried out on the Link48 platform: For primary antibodies raised in mouse, endogenous mouse immunoglobulin was blocked using M.O.M. (Mouse on Mouse) Blocking Reagent (Vector Laboratories) and non-specific interactions blocked using Dako Protein Block (Agilent) prior to application of the primary antibody. Rat primary antibodies were detected using Rat Histofine Reagent (Nichirei Biosciences), rabbit primary antibodies using ImmPRESS HRP Horse anti-Rabbit IgG (Vector Laboratories), and the mouse primary antibody using EnVision anti-mouse-HRP reagent (Agilent). For staining carried out on the Ventana Discovery, slides were deparaffinised using Roche Discovery Wash and HIER was performed for 64 min at 95 °C using CC1 (high pH) buffer. Endogenous peroxidase was blocked using ChromoMap Inhibitor CM (Roche). Primary antibodies were diluted in Ventana Primary Antibody Diluent with casein protein block (Roche) and applied manually. Slides were incubated at room temperature for 32 min before incubation with UltraMap anti-Rabbit HRP secondary multimer for 8 min. Slides were visualised using Roche ChromoMap and haematoxylin counterstain. Slides were digitised using a Nanozoomer XR (Hamamatsu) using a 20X dry objective and the ndpi images imported into QuPath[129] for digital analysis.

## Histopathology analysis

Morphological assessment of essential pathological features of primary and organoid-derived tumours was made on digital scans of H&E sections. Features defining tumour grade (tubule formation, nuclear pleomorphism, mitotic count) were evaluated along guidelines for clinical purposes, as defined by the Royal College of Pathologists (RCPath). Additional characteristics of the tumour and tumour microenvironment, including tumour necrosis, stromal features and immune infiltrates, were also assessed. A semi-quantitative assessment of RIPK1 expression on primary tumours and normal lobules was carried out by applying the H-score method, which incorporates both staining intensity and percentage of stained cells at each intensity level as follows: $\text{H-score} = (0 \times \%\ \text{negative cells}) + (1 \times \%\ \text{weak positive cells}) + (2 \times \%\ \text{moderate positive cells}) + (3 \times \%\ \text{strong positive cells})$, with overall score ranging from 0 to 300. For semi-quantitative assessment of ER and PR expression, the Allred score (0–8 Quick

score) was employed, in keeping with the recommendactions by the RCPath method for clinical practice (score for proportion: 0 = no staining, 1 = < 1% nuclei staining, 2 = 1–10% nuclei staining, 3 = 11–33% nuclei staining, 4 = 34–66% nuclei staining, 5 = 67–100% nuclei staining; score for intensity: 0 = no staining, 1 = weak staining, 2 = moderate staining, 3 = strong staining). Current consensus is that the recommended cut-off point for positivity versus negativity for ER status is ≥1% of tumour cells. For HER2 expression, the semi-quantitative method, which is based on the intensity of staining and percentage of membrane positive cells, recommended by RCPath was employed. This method gives a score range of 0 to 3+, with samples scoring 3+ regarded as positive, and those scoring 0/1+ as negative. Borderline scores (2+) are regarded as equivocal. For the QuPath quantification of stroma to tumour ratio on primary and organoid-derived tumours, H&E stained images were manually annotated with areas of "Stroma" and "Tumour." Whitespace was classified as "*Ignore". These annotations were used to train a pixel classifier using Random Trees at a resolution of 1.81 µm/pixel and utilising all multiscale features. The area occupied by each was used to calculate the stroma to tumour ratio in each specimen. The luminal space was calculated by subtracting the area classified as "Tumour" and the area classified as "Stroma" from the parent annotation area. For the QuPath quantification of infiltration of organoid-derived tumours by F4/80 positive cells, IHC images had their haematoxylin and DAB colour vectors automatically calculated, and the DAB channel was threshold at an Optical Density of 0.1 to create a binary mask. The tumours were manually annotated to include only tumour mass and exclude surrounding tissues. The total area of tumour occupied by positive pixels was calculated. For the QuPath quantification of infiltration of organoid-derived tumours by CD163 or CD8 positive cells, IHC images had their haematoxylin and DAB colour vectors automatically calculated and the stains deconvoluted. The tumour boundary was annotated manually using the polygon annotation tool and surrounding tissues and necrotic regions were excluded. The number of positive cells within the tumour mass (intratumoral area) was quantified using Positive cell detection plug-in. The sum optical density (OD) channel was used for cell detection at a threshold of 0.1. Cells were classified as Positive when the mean DAB OD exceeded 0.07. To count positive cells surrounding the tumour boundary (peritumoral area), the tumour boundary annotation was expanded by 100 µm and 50 µm for CD8 and CD163, respectively. For the QuPath quantification of PD-L1 positive cells, the DAB mean is the average of DAB stain optical density values (at a 2 µm resolution per pixel) across the annotation area (not multiple fields of view). The annotation area for these scans was the area covered by tumour cells, leaving out the surrounding tissue (adipose/lymph node etc).

## Human tumour samples

Human tumour samples used for histological comparison in Fig. 2G consisted of three independent cases (n = 3). Samples were analysed by H&E staining and used for qualitative comparison with mouse tumours to assess histopathological similarity. The samples were derived from female patients diagnosed with triple-negative breast cancer, aged 38–68 years at the time of diagnosis. Samples were obtained from a cohort established at the Institute for Oncology and Radiology of Serbia (IORS), a national comprehensive cancer centre, between 2009 and 2013. The retrospective use of biological material for research purposes was approved by the IORS Ethics Committee (Belgrade; approval date: 10 February 2013). All patients had provided written informed consent for treatment at the time of care. No participant compensation was provided.

## Mice and in vivo treatments

All animal procedures were conducted within the guidelines of UK Home Office in accordance with Animals (Scientific Procedure) Act

(ASPA) 1986, amended 2012 and the institutional guidelines of the Institute of Cancer Research. The Animal Welfare Ethical Review Body (AWERB) reviewed the protocols within the project license. All the animal experiments were conducted in accordance with the Animal Research: Reporting of In vivo Experiments (ARRIVE) guidelines to ensure reproducibility and transparency[109]. Mice were housed in ambient temperatures of 19–23 °C, humidity between 40 and 70% and 12 h light/dark cycles. *Blg-Cre;Brca1^{fl/fl}p53^{+/-}* mice (C57BL/6) were aged until 12 to 15 months, with some mice undergoing at least one round of pregnancy and assessed for spontaneous tumour development in the mammary glands. Mice were culled when the tumours reached an average size of 12 to 15 mm and were processed to generate organoids as described above. Female C57BL/6J, C57BL/6J-*Mlkl^{-/-}* and NSG (*NOD.Cg-Prkdc^{scid} Il2rg^{tm1Wjl}/SzJ*) mice, aged between 6 to 8 weeks were used for tumour studies, including therapeutic interventions (C57BL/6J and C57BL/6J-*Mlkl^{-/-}*). Mice were purchased from Charles River (C57BL/6J and NSG) or obtained by in-house breeding (C57BL/6J-*Mlkl^{-/-}*) and enrolled in the study post an acclimatisation period of at least 1 week. Mice were injected in the 4th right mammary fat pad with $5 \times 10^4$ organoids. Tumour growth was measured twice a week (blinding) using a digital calliper and volumes (in mm³) were calculated with the formula $0.52 \times length \times width \times width$. Mice were treated when tumours reached a volume of 50–100 mm³ and randomised into different treatment groups based on their average tumour volumes. Mice were culled when tumours reached a mean diameter of 15 mm. IAP antagonist ASTX660 (A) (3 mM/injection), Emricasan (E) (1 mM/injection) and diABZI (1 µg/injection) were administered intratumorally (solubilised in Captisol), and αPD-1 was administered intraperitoneally as indicated in Figs. 5, 6 and Supplementary Figs. 6, 7. For the dose optimisation of diABZI, mice were treated intratumorally with diABZI at 1 µg, 5 µg and 10 µg per injection (Supplementary Fig. 8). Based on their treatment responses, mice were categorised under complete response (tumour free for at least 60 days), partial response (tumour volume reduces to <100 mm³) and progression (no response). *p* values were calculated using Fisher's exact test comparing responders (complete and partial) with non-responders. Mice that were tumour free for at least 2 months (cured mice) were rechallenged to assess their immune memory. Naïve and cured mice were injected with $5 \times 10^4$ BP962 organoids in the left 4th mammary fat pad and monitored as described above. To assess early changes in tumours post treatment, mice were given a single injection and tumours harvested post treatment as indicated in Figs. 5 and 6. In Figs. 3 and S3, S4 spleens and tumours with the following tumour sizes, <5 × 5 mm, 6 to 10 mm and 15 × 15 mm, were harvested from mice for histology and immunohistochemical analysis. Tumour growth was monitored throughout the study, and no animal exceeded the ethically permitted tumour size of 15 ×15 mm.

## Statistics and reproducibility
Graphs and statistical analysis were performed using GraphPad Prism v9.5.1. The statistical analysis performed for each data set is described in the corresponding figure legend. Error bars indicate standard deviation (SD) or standard error of the mean (SEM), as indicated. All statistical tests were two-sided, and no statistical methods were used to predetermine sample size. No data were excluded from the analyses unless stated otherwise. Adjustment for multiple comparisons was performed unless indicated in the figure legends.

## Reporting summary
Further information on research design is available in the Nature Portfolio Reporting Summary linked to this article.

## Data availability
All additional raw data, including uncropped scans of all western blots and gels, are publicly available in Zenodo under accession code: https://doi.org/10.5281/zenodo.18130193 (Version 2). All DNA and RNA datasets have been deposited to SRA and are available through accession ID: PRJNA1211523, URL: https://www.ncbi.nlm.nih.gov/sra?term=PRJNA1211523. Source data are provided with this paper.

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

## Acknowledgements

The authors would like to thank Taiho Oncology Inc. (formerly known as Astex Pharmaceuticals Inc.) for their support throughout the project. We would like to express our gratitude to Snezana Susnjar and Natasa Medic Milijic for their contributions to the development and curation of the triple-negative breast cancer dataset at the Serbian Institute of Oncology, which was used in this study to illustrate the similarities between our animal model and human pathology. Further, we would like to thank Jingsong Yang, Jong Yu, Hank Schmidt, Scott Pesiridis, Joshi Ramanjulu, and Michael Adam from GSK for their support during the initial phase of the project. We would like to thank all members of the Meier laboratory for their support and discussions during the work. Work in the Murphy laboratory is funded by National Health and Medical Research Council of Australia grants (1172929 to J.M.M.; 9000719) and the Victorian State Government Operation Infrastructure Support scheme. Work in the Lawrence lab is funded by the City of London CRUK Centre (CTRQQR-2021\100004). Work in the Meier laboratory is funded by Breast Cancer Now as part of Programme Funding to the Breast Cancer Now Toby Robins Research Centre (CTR-QR14-007), CRUK programme funding (C26866/A24399), BBSRC (BB/W017261/1) and Worldwide Cancer Research (23-0146). We acknowledge NHS funding to the NIHR Biomedical Research Centre. This study represents independent research supported by the National Institute for Health Research (NIHR) Biomedical Research Centre at The Royal Marsden NHS Foundation Trust and The Institute of Cancer Research, London. The views expressed are those of the authors and not necessarily those of the NIHR or the Department of Health and Social Care.

## Author contributions

P.M., T.T., and J.C. conceived the study, and J.C., P.M., and T.T. designed the research and wrote the paper. P.M., T.T., J.C., W.F., and A.R. generated the figures. W.F. designed, performed, and analysed in vivo experiments in Figs. 1, 3, 5–7, S2, S4 and S6–S10. J.C. designed, performed, and analysed experiments in Figs. 1–6 and S2–S5. A.R. designed, performed, and analysed experiments in Figs. 7, S5 and S10. R.S. designed, performed, and analysed experiments in Figs. S1 and S2. E.G. designed, performed, and analysed experiments in Fig. S2. T.T. designed, performed, and analysed experiments in Figs. 1, 3, 4, S3, S2, S6–S8 and analysis of the multiplex assays. R.S., T.T., J.C., W.F., and E.G. contributed to the establishment of the organoid sample bank. S.W.J. and S.L. designed, performed, and analysed experiments in Figs. 6, S3 and S9. C.T. designed, performed, and analysed experiments in Fig. S7. J.K., S.J., M.A., and S. Lumbard contributed to the in vivo studies. R.W., N.G., V.B., J.M., M.G., C.S., G.E., N.R., A.B.M., R.A.-E., K.B., F.W., R. Scrimgeour, and D.R. performed and analysed diverse in vitro and histochemical assays. G.W., M.S., and T.S. advised on the in vivo application of the ASTX660 compound. D.K.I. and I.R. provided and analysed the human tumour samples. A.L.S., S.T., S. Lawson, J.M.M., S.H., I.R., D.P.C., A.G., and T.L. supervised specific aspects of the study. W.F., A.R., E.N.A., M.J.S., A.M., T.L., I.R., N.G., and A.N.J.T. provided comments on the paper.

## Competing interests

The authors declare no competing interests.

## Additional information

Winnie Fernando[1,16], Jarama Clucas[1,16] ✉, Alberto Rizzo[1,16], Ramsay Singer[1,16], Emily Goode[1,16], Crescens Tiu[1],
Scott Layzell[1], Joshua Konecnik[1], Rebecca Wilson[1], Sidonie Wicky John[1], Samuel Jouny[1], Naomi Guppy[1],
Victoire Boulat[2,3], Jonathan Mannion[1], Maria Goicoechea[1], Shaun Tan[1], Sam Lawson[1], Chris Starling[1],
Gabrielle Elshtein[1], Nivedita Ravindran[1], Anna B. Montgomery[4], Rosa Andres-Ejarque[4], Mark Allen[5], Steven Lumbard[5],
Fredrik Wallberg[6], Kai Betteridge[1], Ross Scrimgeour[1], David Robertson[1], George Ward[7], Martin Sims[7],
Tomoko Smyth[7], Andre L. Samson[8,9], James M. Murphy[8,9,10], Daniela Kolarevic Ivankovic[11], Esther N. Arwert[1],
Dinis P. Calado[3], Anita Grigoriadis[2,12], Matthew J. Smalley[13], Alan Melcher[1], Syed Haider[1], Toby Lawrence[4,14],
Ioannis Roxanis[1], Andrew N. J. Tutt[1,15], Tencho Tenev[1] ✉ & Pascal Meier[1] ✉

[1]The Breast Cancer Now Toby Robins Research Centre, The Institute of Cancer Research, Fulham Road, London, UK. [2]Cancer Bioinformatics, School of Cancer
& Pharmaceutical Sciences, Faculty of Life Sciences and Medicine, King's College London, London, UK. [3]Immunity and Cancer Laboratory, The Francis Crick
Institute, London, UK. [4]Centre for Inflammation Biology and Cancer Immunology, School of Immunology & Microbial Sciences, King's College London,
London, UK. [5]Biological Services Unit, The Institute of Cancer Research, London, UK. [6]Quell Therapeutics Ltd, Translation & Innovation Hub, London, UK.
[7]Astex Pharmaceuticals, Cambridge, UK. [8]The Walter and Eliza Hall Institute of Medical Research, Parkville, VIC, Australia. [9]Department of Medical Biology,
University of Melbourne, Parkville, VIC, Australia. [10]Drug Discovery Biology, Monash Institute of Pharmaceutical Sciences, Monash University, Parkville,
VIC, Australia. [11]The Royal Marsden NHS Foundation Trust, London, UK. [12]Breast Cancer Now Unit, School of Cancer & Pharmaceutical Sciences, Faculty of Life
Sciences and Medicine, King's College London, London, UK. [13]The European Cancer Stem Cell Research Institute, School of Biosciences, Cardiff University,
Cardiff, UK. [14]Centre d'Immunologie Marseille-Luminy, CNRS, INSERM, Aix Marseille Universite, Marseille, France. [15]Breast Cancer Now Research Unit, King's
College London, London, UK. [16]These authors contributed equally: Winnie Fernando, Jarama Clucas, Alberto Rizzo, Ramsay Singer, Emily Goode.
✉e-mail: jarama.clucas@icr.ac.uk; tencho.tenev@icr.ac.uk; pascal.meier@icr.ac.uk

