## [Transparent Peer Review file · Nature Communications]

Necroptosis in Both Tumour and Stromal Compartments Determines Responsiveness to Immunogenic Cell Death-based Immunotherapy

Corresponding Author: Professor Pascal Meier

Version 0:

Reviewer comments:

Reviewer #1

(Remarks to the Author)

This manuscript by Clucas et al focused on the potential role necroptotic death may have in enhancing response of breast cancer to immunotherapy via checkpoint blockade. They developed murine breast cancer models to mimic human TNBC. Half of the manuscript was devoted to the characterization of their breast cancer organoids. They studied three of these organoid lines, one of which is immunologically hot whereas the other two were considered to be immunologically cold. These are very interesting models that can be used to study the response of breast cancers to immunotherapy. Overall, these are well conducted studies, but this reviewer has some concerns that should be addressed.

Major concerns:

1. The main conclusion is that the BP962 organoid line when treated with a regimen to shift the response in tumor cells to necroptosis, there was better response to anti-PD1. While the regimen used is known to lead to necroptosis, the conclusion will be stronger if the authors can show a couple of things.
 - A. If they use Mkl1^{-/-} BP962 tumors, the enhanced response to anti-PD1 is lost. This will show that it is necroptosis in the tumor cells and not in the stromal cells that is having this effect. It is possible that AXT660/E is causing necroptosis in the stromal cells in the TME, and this may have some role. They were able to delete Mkl1 in the 903 line. They should do this with the 962 line and conduct the same treatment experiments as in Fig 5e-g.
 - B. Show that necroptosis is occurring within the tumor tissue for e.g., by staining tumor sections with anti-phospho-MLKL.
2. Why are the organoids and E0771 not dying with TNF+AXT660? They should be undergoing RDA (Ripk1-dependent apoptosis). Is this due to the assay used, which is PI exclusion? Would authors see cleaved Casp8 or cleaved Casp3 if they perform western blots? ASTX660 appears to have some effect in combo with anti-PD1 in Fig 5C.
3. Do the cold tumors, 903 and 487, benefit from combination of STING agonist, ASTX660, Emricasan and anti-PD1?

Minor concerns:

4. Authors kept referring to using Pembrolizumab in the text on page 13. They are not using pembrolizumab, which is specific for human PD1. They are using anti-mouse PD1 clone RMP1-14. This should be changed.
5. Is TNF coming from the tumor or from the host immune cells that is driving the response? Because the tumors are able to die in vitro with just AXT660 and emricasan, they are likely producing TNF? Authors should discuss this.
6. 903 and 487 can undergo necroptosis and seem to be better at undergoing necroptosis than 962 (Fig 4e), yet they don't respond in vivo (Fig S6 c,f). This would suggest that necroptosis is not a determinant of the response to anti-PD1. Something else needs to also be present. Authors should temper their conclusions a bit – necroptosis is only helpful if other conditions are already in place. Something else is already present to make that tumor 'hot' to begin with and provoking necroptosis is useful only when those conditions are in place. This would also fit with their data in Fig 6 with the STING agonist.

Reviewer #2

(Remarks to the Author)

Clucas J, Fernando W and colleagues investigate the significance of necroptosis-mediated immune modulation in enhancing immunotherapy in triple-negative breast cancer. Authors use BLG-Cre; Brca1f/tp53+/- mouse model derived tumors to construct organoids. Subsequently, the authors validated that the organoids derived from different mice resembled human triple-negative breast cancer (TNBC) but also exhibited significant heterogeneity. The organoids were re-implanted into mice, stably recapitulating the histological and microenvironmental characteristics of the primary tumor. The authors constructed three mice cohorts for efficacy testing and analysis by selecting three organoid lines, two of which were inclined towards "cold tumors" and one towards "hot tumors". The authors validated that ASTX660-mediated immunogenic cell death enhanced the efficacy of immunotherapy in the "hot tumor" mice cohort, while it had limited effect on the "cold tumor" cohort. For "cold tumors," the authors demonstrated that combining the induction of immunogenic cell death with the activation of innate immunity through a STING agonist was necessary to achieve significant therapeutic efficacy. Overall, the authors try to validate the known antitumor effects of necrotic apoptosis combined with immunotherapy by constructing mice cohorts. But my interest in this paper was limited by the following concerns.

1. This paper is not innovative enough. The ability of BLG-Cre; Brca1f/tp53+/- mouse model to produce a heterogeneous but similar phenotype to human basal-type breast cancer has been demonstrated (PMID: 20804975). Moreover, the ability to stably reproduce the characteristics of primary tumors through organoid transfer in mice has been reported as a well-established method (PMID: 32690957). And the enhancing effect of necroptosis on immunotherapy, including the combination regimen of ASTX660 and immunotherapy used in the text, has also been reported (PMID: 31900279, PMID: 30393585 and PMID: 39560995). The "hot tumors" identified in the study lacks a positive control.
2. The article's logic needs to be improved. Fig. 3J lacks strong relevance to the surrounding context, making it feel abrupt to the reader. Additionally, the introduction of the STING pathway seems forced, with no data to support the connection.
3. The detection of CSF1 in Fig. 4D seems selective. There are many reasons for tumor-induced spleen enlargement; was there a thorough exclusion process to confirm that CSF1 is the primary cause?
4. The abbreviations in the figures, such as SM and A/E, should be defined in the figure legend or in the main text.
5. In the analysis of the therapeutic effects of ASTX660/pembrolizumab on the "cold" and "hot" tumor cohorts, it seems that the lack of statistical significance in "cold tumors" cohorts is due to the different number of mice in each cohort.
6. The rigor of the conclusions stated in the article needs to be improved. For example, the sentence "However, animals with less immune infiltrated tumors do not benefit from such treatments, even under necroptotic settings and double immune checkpoint blockade" should be revised. The conclusion should state that there is some benefit, but it is not significant enough. The sentence "First, we evaluated the ability of diABZI to drive STING signaling in BP903 and BP487 organoids in vitro. While treatment with diABZI readily triggered activation of IRF3 and interferon signaling (STAT1 activation) in both organoids, the STING agonist showed little cell death activity in vitro, alone or in combination with IAP antagonism" also needs a more nuanced statement.
7. For Fig. 6E and F, including a group with "A/E" would allow for a better comparison of the antitumor effects of the combination of diABZI and "A/E."
8. In Fig. 6O, the improvement in antitumor function should be assessed using flow cytometry to detect IFN- γ and perforin, which is better than measuring their mRNA levels.

Reviewer #3

(Remarks to the Author)

Summary of noteworthy results:

This study established tumor organoid lines from Brca1/p53 mouse models of human triple-negative breast cancer and generated orthotopic mammary tumors in immunocompetent mice.

Using this as a platform, they identified one tumor organoid line with both high immune infiltration and high expression of RIPK1, a mediator of apoptosis/necroptosis cell death, and used it to demonstrate the efficacy of RIPK1-induced immunogenic cell death. Specifically, double treatment with a cell-death-inducing agent (ASTX660, an IAP antagonist) and anti-PD1 (pembrolizumab) delayed BP962 tumor progression, whereas triple treatment (adding emricasan, a pan-caspase inhibitor) to promote necroptosis over apoptosis further enhanced the anti-tumor effect, leading to complete cure of the mice and immune memory against subsequent re-inoculation of tumor cells. The treated tumors showed increased T cell and decreased macrophage infiltration. In contrast, the same anti-tumor effect was not seen in tumors induced from two other organoid lines with lower RIPK1 levels and less immune infiltration. The authors therefore proposed RIPK1 as a biomarker for necroptosis-based enhancement of immune checkpoint blockade therapy in immune-infiltrated tumors.

The authors further showed that immune-cold tumors (derived from two organoid lines, BP903 and BP487) can be sensitized to necroptosis-induced cell death (ASTX660/emricasan) by cGAS-STING activation (via diABZI) that activates immune infiltration through viral mimicry. The treatment reshaped the tumor immune microenvironment with more effector lymphocytes and fewer immunosuppressive cells.

Overall assessment:

This study provides interesting and exciting findings that suggest a new potential approach to enhance immunotherapy efficacy for patients with triple-negative breast cancer. This is not the first study to use Brca1/p53-mutant mammary tumor organoids and orthotopic transplantation to study drug response and resistance (see

<http://pubmed.ncbi.nlm.nih.gov/29256493/> for example). However, this study used these organoids in an immunocompetent host setting, and the findings of the anti-tumor effect of necroptosis in combination with immune-modulating treatment are novel and promising.

The methodology, analysis, and interpretation are sound. However, more data are needed to strengthen the current conclusion, which is drawn based on the comparison between only one single tumor organoid line (BP962, with high RIPK1 and high immune infiltration) versus two other lines (RIPK1-low and/or poor immune infiltration). The experiments should be repeated on additional tumor organoid line(s) with a similar profile to BP962 to establish that the findings are generalizable.

Major comments:

- Does other tumor organoid line(s) with similar RIPK1 expression and immune infiltration to BP962 show the same sensitivity to the combination treatment(s) using ASTX660/pembrolizumab or ASTX660/pembrolizumab/E treatment?
- Is RIPK1 necessary and sufficient for the anti-tumor effects of ASTX660/pembrolizumab or ASTX660/pembrolizumab/E treatment in immune-infiltrated tumors? For example, does knockdown/knockout of RIPK1 in a RIPK1-high tumor organoid line (e.g. BP962) abolish the anti-tumor effects of such combo treatment(s)? Alternatively, can overexpression of RIPK1 in RIPK1-low/immune high tumors enable the anti-tumor effect of such combination treatment(s)?
- What drives the heterogenous expression of RIPK1 in these primary tumors and their derivative tumor organoid lines? Can the authors provide data to suggest a potential mechanism, or at least discuss some speculation in the text?
- It would make the paper even stronger if the authors could provide complementary data from humans to reinforce at least some of their conclusions drawn in mouse models.

Minor points:

- "Notably, the individual organoid lines remained stable, and their respective TOs were highly reproducible within each line, as evidenced by mRNA analysis and mutational profiling (Fig. 2e,f)". Figure 2f showed CNAs present in primary tumors and absent in organoids and vice versa, potentially due to the expansion of clonal subpopulation and/or acquired CNA in culture. The authors should modify their language about stability and reproducibility to reflect these differences between primary tumors and organoid lines.
- Figure 2h: Scale bars representing different sizes should be more clearly labeled.
- Figure 3a-b: Are the "normal mammary glands" analyzed after the lymph nodes have been removed? The author should clarify in the figure legend and/or methods section.
- Figure 3j: genes should be labeled in the heatmaps.
- Figure S5c schematic indicates that cells were stained with PI and Hoechst, whereas the data panels (Fig. 4e,g,h,i, and S5d) show PI and DAPI. Please clarify.

Version 1:

Reviewer comments:

Reviewer #1

(Remarks to the Author)

This revised manuscript by Clucas et al has undergone significant revision with the inclusion of several pieces of additional data. They have done an admirable job addressing the concern of this reviewer. A major takeaway is that despite these tumors all arising from the same oncogenic drivers in a mouse model, they developed into a heterogenous group of tumors with different behaviour to immune/pharmacological interventions. It further illustrates the challenge of treating human patients where the genetic diversity is even greater. This reviewer has no further concerns.

Reviewer #2

(Remarks to the Author)

The authors have comprehensively addressed all my previous concerns through extensive revisions, including strengthened experimental design and clarified textual. The incorporation of substantial new data has convincingly supported their original conclusions. Importantly, the study now presents novel insights that significantly elevate the overall impact and novelty of the work. I have no remaining issues and support its publication in its current form.

Reviewer #3

(Remarks to the Author)

The authors have significantly improved their manuscript and have addressed all the concerns I raised in the first round of

review.

Nature Communications manuscript NCOMMS-25-01910

We would like to thank you and the referees for the constructive and helpful comments on our manuscript. We were delighted to see that the reviewers thought that our study was well performed.

We have followed the reviewers' suggestions and have considerably expanded our manuscript with extensive new experiments (using an additional 430 animals), which have corroborated our existing data and **significantly strengthened** our ms. The new data includes **53 new panels**, which we have incorporated into our response to reviewer's comments for ease of access. The data of these additional experiments have been fully embedded into our revised ms. Additional new panels have been framed with **red** boxes in the figures of our revised manuscript, for ease of access.

Overall, our new data have addressed the following points:

1. We provide new evidence demonstrating that necroptosis enhances treatment responses both within tumour cells (BP962^{Ripk1-KO} and BP962^{Mik1-KO}) and in the tumour microenvironment (*Mik1*^{-/-} mice) (**new Fig. 7 and Suppl. Fig. S10**).
2. We now show that necroptosis indeed occurs within the tumour tissue (**new Suppl. Fig. S6A**).
3. We clarify why cells do not undergo death in response to TNF/ASTX660 treatment (**new Suppl. Fig. S5E–H**).
4. We provide additional data evaluating whether “cold” tumours benefit from combined STING agonist/ASTX660/emricasan and anti-PD-1 treatment (**new Suppl. Fig. S8D–L**).
5. We introduce an additional tumour model that closely recapitulates the BP962 phenotype (**new Suppl. Fig. S6B–G**).
6. We increased animal numbers in cohorts with weaker phenotypes to assess statistical significance (**new Suppl. Fig. S7A–G**).
7. We incorporated additional controls (**new Fig. 6D–J**) and extended immune-cell characterisation using FACS analysis (**new Fig. 6O and Suppl. Fig. S9**).
8. We addressed all remaining reviewer comments, revising the text for clarity and precision throughout. We further emphasise and clarify the novelty and conceptual advances of our study.

Please find below a point-by-point response to the reviewers' comments, with the **reviewers' comments in blue boxes** and our response in 'plain text'.

Further, we are attaching a private, Reviewer-Only link to all the DNA and RNA data sets:
<https://dataview.ncbi.nlm.nih.gov/object/PRJNA1211523?reviewer=fjv9lauhkbtjablad5pvegh6c5>

REVIEWER COMMENTS

Reviewer #1 (Remarks to the Author):

This manuscript by Clucas et al focused on the potential role necroptotic death may have in enhancing response of breast cancer to immunotherapy via checkpoint blockade. They developed murine breast cancer models to mimic human TNBC. Half of the manuscript was devoted to the characterization of their breast cancer organoids. They studied three of these organoid lines, one of which is immunologically hot whereas the other two were considered to be immunologically cold. These are very interesting models that can be used to study the response of breast cancers to immunotherapy. Overall, these are well conducted studies, but this reviewer has some concerns that should be addressed.

Major concerns:

1. The main conclusion is that the BP962 organoid line when treated with a regimen to shift the response in tumor cells to necroptosis, there was better response to anti-PD1. While the regimen used is known to lead to necroptosis, the conclusion will be stronger if the authors can show a couple of things.

A. If they use *Mkl1*^{-/-} BP962 tumors, the enhanced response to anti-PD1 is lost. This will show that it is necroptosis in the tumor cells and not in the stromal cells that is having this effect. It is possible that AXT660/E is causing necroptosis in the stromal cells in the TME, and this may have some role. They were able to delete *Mkl1* in the 903 line. They should do this with the 962 line and conduct the same treatment experiments as in Fig 5e-g.

B. Show that necroptosis is occurring within the tumor tissue for e.g., by staining tumor sections with anti-phospho-MLKL.

1A: The role of *Mkl1* in tumour cells vs stroma.

This is an excellent point, which we have now addressed comprehensively in the revised manuscript with an additional new Figure (new Fig. 7).

We fully agree that elucidating how immunogenic cell death is initiated within the tumour, and determining whether this response originates from cancer cells or stromal components, is of key mechanistic importance.

To directly address this, we implemented a two-pronged experimental strategy (new Fig. 7 and Suppl. Fig. S10):

1. Tumour: We generated a BP962 tumour organoid line deficient in MLKL (BP962^{*Mkl1*-KO}) and compared its treatment response to that of isogenic controls (new Fig. 7D-G). Additionally, we also assessed the role of RIPK1 (new Fig. 7A-C).

2. Stroma: We evaluated the treatment effect *in vivo* using recipient mice that were either wild-type (WT) or *Mkl1*^{-/-} (new Fig. 7H,I).

1. BP962^{*Mkl1*-KO}: To directly test the requirement of MLKL in BP962 cancer cells, we generated BP962^{*Mkl1*-KO} organoid lines using Cas9/CRISPR-mediated genome editing. Successful deletion of MLKL protein and resulting resistance to necroptosis were confirmed by Western blot and functional analysis (new Suppl. Fig. S10A,B), attached below. Compared with parental BP962 organoids, BP962^{*Mkl1*-KO} cells formed tumours at a slightly reduced growth rate in the absence of treatment, resulting in delayed mortality in tumour-bearing mice (new Suppl. Fig. S10C,D). Nevertheless, upon treatment, BP962^{*Mkl1*-KO} tumours significantly responded to ASTX660/EC. However, addition of α PD-1 did not provide any further survival benefit beyond ASTX660/EC alone (new Fig. 7E-G). This contrasts with necroptosis-

proficient BP962 tumours, where α PD-1 co-treatment enhances the survival benefit of ASTX660/EC (Fig. 5E-G), suggesting that MLKL-dependent necroptosis contributes to the immunological potentiation of this combination therapy.

2. The role of stromal cell necroptosis: To assess the contribution of stromal necroptosis to therapy response, BP962 organoids were transplanted into *Mkl1*^{-/-} recipient mice. While ASTX660/EC/ α PD-1 treatment produced a measurable effect, its efficacy was significantly reduced in *Mkl1*^{-/-} hosts compared to WT animals, with no surviving animals observed in the *Mkl1*^{-/-} group (compare **new Figure 7H,I** with previous Fig. 5E-G (WT host)).

Together, these findings indicate that necroptosis within stromal cells plays a critical role in mediating the therapeutic effect. In addition, necroptosis of tumour cells, and the release of tumour-associated antigens and DAMPs, appears essential for a response to α PD-1. Therefore, necroptosis of both stromal and tumour compartments is required to elicit a fully effective anti-tumour immune response.

New Fig. 7. Cancer cell intrinsic and tumour micro-environmental necroptosis contributes to the anti-tumour effect. (A) Tumour growth curves of BP962 *Ripk1*^{KO} in C57BL/6J mice: Control (n = 8), A/E (n=6) and A/E/ α PD-1 (n=9). Each line represents one animal and thick lines represent average tumour growth. Curves represent one independent experiment. **(B)** Kaplan-Meier survival curves of BP962^{*Ripk1*-KO} T⁰ tumour-bearing animals. **(C)** Pie

charts depicting the proportion of mice, which progressed, partially responded, or fully responded to the indicated treatments. **(D)** Schematic representation depicting the experimental setup to evaluate the contribution of tumour versus stromal necroptosis. The treatment regimen is depicted on the right. **(E)** Tumour growth curves of BP962^{Mikl-KO} T⁰s: Control (n = 8), A/E (n = 12) and A/E/αPD-1 (n = 12). Each line represents one animal and thick lines represent average tumour growth. Curves represent one independent experiment. **(F)** Kaplan-Meier survival curves of BP962^{Mikl-KO} tumour-bearing animals, treated as in (D). **(G)** Pie charts depicting the proportion of mice from (F), which progressed, partially responded, or fully responded to the indicated treatments. **(H)** Tumour growth curves of BP962 T⁰s grown in *Mikl*^{-/-} animals (see Fig. 5E-G for WT control). Treatments: Control (n = 6) and A/E/αPD-1 (n = 9). Each line represents one animal and thick lines represent average tumour growth. Curves represent two independent experiments. **(I)** Kaplan-Meier survival curves of BP962 tumour-bearing *Mikl*-KO animals, treated as in (D). A=ASTX660; E=emricasan;

1B. Show that necroptosis is occurring within the tumor tissue for e.g., by staining tumor sections with anti-phospho-MLKL.

We followed the reviewer's suggestion and attempted to detect p-MLKL in tumour tissue. To this end, we analysed tumour lysates from treated (n = 6) and untreated (m = 4) mice by western blotting. As shown in new Suppl. Fig. S6A, clear induction of p-RIPK1 and p-MLKL was detected 24 hours after ASTX660/EC treatment, confirming necroptosis activation in the tumour. Together with the data shown in new Fig. 7, these findings support the conclusion that ASTX660/EC indeed induces necroptosis.

We also attempted to detect p-MLKL in tissue sections by IHC. However, despite extensive optimisation to make the anti-phospho-MLKL antibody (CST37333) work in tissue sections by IHC, we were unfortunately unable to detect a specific signal in control samples.

2. Why are the organoids and E0771 not dying with TNF+AST660? They should be undergoing RDA (Ripk1-dependent apoptosis). Is this due to the assay used, which is PI exclusion? Would authors see cleaved Casp8 or cleaved Casp3 if they perform western blots? ASTX660 appears to have some effect in combo with anti-PD1 in Fig 5C.

We thank the reviewer for raising this point.

We have addressed this experimentally by performing complementary viability and cell death assays. During the characterisation of the organoids, we carried out **CellTiter-Glo (CTG)** assays to assess viability, as well as **ImageXpress/Celigo (PI/Hoechst)** assays to evaluate cell death.

Treatment of organoids with ASTX660, either alone or in combination with TNF, did not result in a significant reduction in viability. However, as the CTG assay has limited sensitivity, we cannot exclude the possibility that low-level apoptosis occurred but remained below the detection threshold of PI-based methods. In addition, it is worth noting that caspase-8/cFLIP heterodimers can suppress RIPK1-dependent cell death downstream of IAP antagonists by cleaving RIPK1, which could contribute to the limited cytotoxic response observed under these conditions. Accordingly, western blot analysis of treated organoids indicates that TNF/ASTX660 treatment indeed results in detectable caspase-8 activation and RIPK1 cleavage (**new Suppl. Fig. S5E-H**). This could either result in low level apoptosis induction or caspase-8-mediated suppression of RIPK1-dependent apoptosis.

3. Do the cold tumors, 903 and 487, benefit from combination of STING agonist, ASTX660, Emricasan and anti-PD1?

We have expanded our experiments to treatment combination.

Interestingly, while STING agonist treatment “heated up” BP903 and BP487 tumours and enhanced the anti-tumour efficacy of ASTX660/EC, the addition of α PD-1 did not confer further therapeutic benefit (New Suppl. Fig. S8). These findings suggest that, although STING activation effectively increases tumour kill, this enhancement alone is insufficient to sustain or amplify the immune response when combined with α PD-1 therapy.

New Suppl. Fig. S8D-L. Effect of α PD-1 on A/E/STING treatment. (D) Schematic representation depicting the treatment regimen. (E) Tumour growth curves of BP903 T⁰s: Control (n = 7), diABZI/E (n = 8), diABZI/ α PD-1 (n = 8), diABZI/A/E (n = 10) and diABZI/A/E/ α PD-1 (n = 10). Each line represents one animal and thick lines represent average tumour growth. Curves represent one independent experiment. (F) Tumour growth kinetics (days 0-28) of mice treated as in (D), measured by the area under the curve (AUC). Each point represents the AUC of individual mice from (E). (G) Kaplan-Meier survival curves of BP903 T⁰ tumour-bearing animals, treated as in (D). (H) Pie charts depicting the proportion of mice from (B), which progressed, partially responded, or fully responded to the indicated treatments. (I) Tumour growth curves of BP487 T⁰s: Control (n = 9), diABZI/A/E (n = 9) and diABZI/A/E/ α PD-1 (n = 9). Each line represents one animal and thick lines represent average tumour growth. Curves represent one independent experiment. (J) Tumour growth kinetics (days 0-25) of mice treated as in (D), measured by the area under the curve (AUC). Each point represents the AUC of individual mice from (F). (K) Kaplan-Meier survival curves of BP487 T⁰ tumour-bearing animals, treated as in (D). (L) Pie charts depicting the proportion of mice from (K), which progressed, partially responded, or fully responded to the indicated treatments.

Minor concerns:

4. Authors kept referring to using Pembrolizumab in the text on page 13. They are not using pembrolizumab, which is specific for human PD1. They are using anti-mouse PD1 clone RMP1-14. This should be changed.

We apologise for this oversight. This has been corrected accordingly.

5. Is TNF coming from the tumor or from the host immune cells that is driving the response? Because the tumors are able to die *in vitro* with just AXT660 and emricasan, they are likely producing TNF? Authors should discuss this.

We have expanded our revised ms to discuss the potential source of TNF.

We now state in the discussion:

Our data raise an important question regarding the cellular origin of TNF that fuels the anti-tumour response upon IAP inhibition. *In vitro*, BP962 tumour organoids undergo cell death when treated with ASTX660 and emricasan, even in the absence of immune cells, suggesting that tumour cells themselves can produce TNF in an autocrine manner. This is consistent with previous reports demonstrating that IAP antagonists can trigger tumour-intrinsic TNF production, creating a self-amplifying death loop through RIPK1 activation. However, *in vivo*, immune-derived TNF is likely to play an additional and significant role. CD8⁺ T cells and myeloid cells are well-established sources of TNF in the tumour microenvironment, particularly after checkpoint blockade. Given that α PD-1 enhances T-cell activation, the therapeutic effect of ASTX660/ α PD-1 is likely reinforced by TNF derived from infiltrating immune cells. The observation that α PD-1 improves efficacy only in necroptosis-competent tumours further supports the notion that immune-derived TNF amplifies inflammatory cell death and cross-priming *in vivo*. Together, these findings suggest a dual-source model in which tumour-intrinsic TNF may initiate RIPK1-dependent immunogenic cell death, while immune-cell-derived TNF further amplifies this response within the tumour microenvironment. Future studies using cell-type-specific TNF perturbation will be required to dissect the relative contribution of tumour- versus immune-derived TNF to treatment efficacy.

6. 903 and 487 can undergo necroptosis and seem to be better at undergoing necroptosis than 962 (Fig 4e), yet they don't respond *in vivo* (Fig S6 c,f). This would suggest that necroptosis is not a determinant of the response to anti-PD1. Something else needs to also be present. Authors should temper their conclusions a bit – necroptosis is only helpful if other conditions are already in place. Something else is already present to make that tumor 'hot' to begin with and provoking necroptosis is useful only when those conditions are in place. This would also fit with their data in Fig 6 with the STING agonist.

We fully agree with the reviewer's insightful comment. This is indeed an important point, and we are actively investigating how stromal and tumour compartments communicate to establish a tumour-specific microenvironment.

Interestingly, we observed a strong correlation between RIPK1 expression levels **and the** composition of the tumour microenvironment. BP962-derived tumours exhibit high RIPK1 expression, whereas BP487 and BP903 tumours express much lower levels. However, in the corresponding organoids *in vitro*, all three lines display comparable RIPK1 expression, indicating that these differences arise only after engraftment and tumour formation *in vivo*.

These findings suggest that the reconstituted tumours faithfully recapitulate key features of the original "patient" tumours. Although the precise factors driving this adaptation remain unclear, it is evident that the stroma is remodelled in a tumour-intrinsic manner according to the organoid's genetic and epigenetic profile. BP962 tumours develop an immune-rich microenvironment with dense macrophage infiltration and direct macrophage-tumour interactions, whereas BP903 and BP487 represent "cold" tumour phenotypes, with macrophages restricted to connective tissue regions or "tumour highways."

Importantly, loss of key regulatory control points, such as through inactivation of IAPs or caspase-8 activity, can prime cancer cells for necroptosis. However, in the absence of a microenvironment rich in TNF and interferon, necroptosis cannot be effectively engaged. This suggests that additional factors,

such as the availability of cytokine ligands and interferon-driven upregulation of MLKL, which sensitises cells to TNF-mediated cytotoxicity, are also required.

Our data further support the notion that stromal necroptosis contributes more prominently to the anti-tumour response than tumour-cell necroptosis alone, although both are required. As macrophages are the principal source of TNF within the tumour, their abundance and spatial distribution likely determine the extent of necroptosis activation and, consequently, responsiveness to immune checkpoint inhibition (ICI).

Consistent with this, the **E0771 model** closely mirrors the BP962 phenotype (see reviewer 3, point 1 below), displaying a similarly macrophage-rich microenvironment and responding even more robustly to combined ASTX/EC/ α PD-1 therapy—likely reflecting its greater susceptibility to necroptosis.

We have expanded our manuscript to discuss this point.

Reviewer #2 (Remarks to the Author):

Clucas J, Fernando W and colleagues investigate the significance of necroptosis-mediated immune modulation in enhancing immunotherapy in triple-negative breast cancer. Authors use BLG-Cre; *Brca1f/fp53+/-* mouse model derived tumors to construct organoids. Subsequently, the authors validated that the organoids derived from different mice resembled human triple-negative breast cancer (TNBC) but also exhibited significant heterogeneity. The organoids were re-implanted into mice, stably recapitulating the histological and microenvironmental characteristics of the primary tumor. The authors constructed three mice cohorts for efficacy testing and analysis by selecting three organoid lines, two of which were inclined towards "cold tumors" and one towards "hot tumors". The authors validated that ASTX660-mediated immunogenic cell death enhanced the efficacy of immunotherapy in the "hot tumor" mice cohort, while it had limited effect on the "cold tumor" cohort. For "cold tumors," the authors demonstrated that combining the induction of immunogenic cell death with the activation of innate immunity through a STING agonist was necessary to achieve significant therapeutic efficacy.

Overall, the authors try to validate the known antitumor effects of necrotic apoptosis combined with immunotherapy by constructing mice cohorts. But my interest in this paper was limited by the following concerns.

1. This paper is not innovative enough. The ability of BLG-Cre; *Brca1f/fp53+/-* mouse model to produce a heterogeneous but similar phenotype to human basal-type breast cancer has been demonstrated (PMID: 20804975). Moreover, the ability to stably reproduce the characteristics of primary tumors through organoid transfer in mice has been reported as a well-established method (PMID: 32690957). And the enhancing effect of necroptosis on immunotherapy, including the combination regimen of ASTX660 and immunotherapy used in the text, has also been reported (PMID: 31900279, PMID: 30393585 and PMID: 39560995). The "hot tumors" identified in the study lacks a positive control.

We thank the reviewer for their constructive feedback and believe that by addressing all comments by all reviewers, we have substantially improved our manuscript and clarified the novelty of our work.

We agree that both the *BLG-Cre; Brca1f/f; p53+/-* model (PMID: 20804975) and organoid-based tumour reconstitution methods (PMID: 32690957) are established in the field. However, the novelty of our study lies in integrating these systems to generate a **diverse organoid biobank** from *Brca1/p53*-deficient tumours and in functionally linking tumour-intrinsic necroptotic susceptibility to the **composition and immune activation state of the resulting tumour microenvironment *in vivo***.

Our project was initiated with the goal of improving therapeutic strategies of breast cancer patients. A major limitation in the field is that much of the current data, including that in breast cancer research, relies heavily on established cell lines that do not accurately recapitulate the complexity of primary tumours. In Molyneux et al., 2010 (PMID: 20804975), the authors demonstrated that *Brca1*-associated basal-like breast cancers arise from luminal progenitors rather than basal stem cells. Loss of *Brca1/p53* in these cells produced tumours that phenotypically copy human *Brca1/p53*-deficient cancers. Building on these findings, our aim was to establish organoid lines from *Brca1/p53*-deficient tumours that more faithfully model human breast cancer. Further our aim was to develop a corresponding ***in vivo* "mouse clinic" platform** to address heterogeneity and explore therapeutic vulnerabilities.

We worked closely with Matt Smalley, a co-author of our study, who originally developed the *Brca1/p53* mouse model. Notably, organoid lines were not established from these tumours in the original study (PMID: 20804975). Thus, our work represents the first systemic derivation and characterization of such organoids.

The development of organoid protocols from diverse tumour types has transformed the field, bringing preclinical models closer to human disease complexity. The study by Padmanaban et al., 2020 (PMID:

32690957) provided a valuable step-by-step protocol for isolating organoids from murine and human tumours, preserving tumour heterogeneity and enabling studies of tumour biology and drug responses. We adapted and refined such approaches to generate organoids from our *Brca1/p53* model. Unlike MMTV-PyMT mouse mammary tumour model used by Padmanaban and colleagues, which has been criticized for its limited resemblance of human breast cancer, our *Brca1/p53* system targets tumour suppressor loss and recapitulate the genetic and phenotypic diversity observed in human *Brca1*-deficient tumours. This inherent heterogeneity poses technical challenges for organoid isolation and maintenance, which we described in detail to provide a resource for other researchers.

Our goal was both to describe this complexity and to investigate the mechanism that drives immunogenic cell death in tumours derived from these organoids, elucidating why certain tumours respond to necroptotic stimuli while other do not.

Previous studies using IAP antagonists such ASTX660 (PMID: 31900279, PMID: 30393585) have shown promising antitumour results. The first-in-human study of ASTX660 (PMID: 31900279) established safety, pharmacokinetics, and cIAP1 target engagement across heterogeneous human cancers. We found no discrepancies with these findings. Indeed, we are currently conducting phase I clinical trial combining ASTX660 with pembrolizumab in patients with breast and other cancers, with data now submitted for publication. Such clinical investigations are essential to advancing this therapeutic strategy in the complex context of human disease.

In our manuscript, we demonstrate that inducing necroptosis in genetically and phenotypically heterogeneous breast tumours does not uniformly result in an immunogenic response. Using genetic models, we show that the outcome of necroptotic signalling is largely determined by the tumour microenvironment that is driven by the tumour cells. Identifying which tumours are likely to respond to necroptotic therapies is therefore critical for translating this approach into clinical benefit, and to our knowledge, has not yet been addressed elsewhere.

As the reviewer correctly noted, necroptosis induction in cancer has been previously demonstrated. However, several of these studies-including the above mentioned PMID: 39560995-employed transgenic systems in which RIPK3 activation (often driven by artificial RIPK3 oligomerization and not pathway driven) within cancer cells directly triggers necroptosis. While this approach effectively induces necroptosis, we believe that true immunogenicity requires engagement of upstream innate immune receptors to induce DAMPs to initiate a productive antitumour immune response.

Furthermore, our data reveal that effective therapy must target not only tumour cells but also the surrounding stroma, as stromal necroptosis plays a pivotal role in shaping the overall immune response.

2. The article's logic needs to be improved. Fig. 3J lacks strong relevance to the surrounding context, making it feel abrupt to the reader. Additionally, the introduction of the STING pathway seems forced, with no data to support the connection.

We thank the reviewer for this comment and have revised the text to clarify the logic linking Fig. 3J to the surrounding context. As demonstrated in our rebuttal, the **stromal compartment** is a key determinant of the response to necroptosis-based therapy. Figure 3 now provides a more coherent framework for interpreting this complexity.

Specifically, **Figure 3J** depicts the **transcriptional activation of NF- κ B and IFN pathways** across three tumour types. These signatures are known to correlate with improved responsiveness to immune-checkpoint blockade-based therapies as both these signalling pathways influence bystander killing of tumour cells. Accordingly, elevated NF- κ B and IFN pathways correlates with improved responses to necroptosis and we believe that this observation represents an important component of the overall mechanism.

The inclusion of the **STING pathway** was not arbitrary but mechanistically driven. Its activation is known to enhance NF- κ B and type I IFN signalling, pathways already implicated by our data (Fig. 3J). However, as shown in our additional experiments (**new Suppl. Fig. S8**), STING agonism did not generate a durable immune response when combined with α PD-1, suggesting that additional contextual factors, such as the spatial distribution of macrophages, are essential for achieving sustained tumour control.

Together, these clarifications strengthen the logical connection between Fig. 3J and the broader narrative of our study.

New Suppl. Fig. S8D-L. Effect of α PD-1 on diABZI/A/E treatment. (D) Schematic representation depicting the treatment regimen. (E) Tumour growth curves of BP903 T⁰s: Control (n = 7), diABZI/E (n = 8), diABZI/ α PD-1 (n = 8), diABZI/A/E (n = 10) and diABZI/A/E/ α PD-1 (n = 10). Each line represents one animal and thick lines represent average tumour growth. Curves represent one independent experiment. (F) Tumour growth kinetics (days 0-28) of mice treated as in (D), measured by the area under the curve (AUC). Each point represents the AUC of individual mice from (E). (G) Kaplan-Meier survival curves of BP903 T⁰ tumour-bearing animals, treated as in (D). (H) Pie charts depicting the proportion of mice from (E), which progressed, partially responded, or fully responded to the indicated treatments. (I) Tumour growth curves of BP487 T⁰s: Control (n = 9), diABZI/A/E (n = 9) and diABZI/A/E/ α PD-1 (n = 9). Each line represents one animal and thick lines represent average tumour growth. Curves represent one independent experiment. (J) Tumour growth kinetics (days 0-25) of mice treated as in (D), measured by the area under the curve (AUC). Each point represents the AUC of individual mice from (I). (K) Kaplan-Meier survival curves of BP487 T⁰ tumour-bearing animals, treated as in (D). (L) Pie charts depicting the proportion of mice from (K), which progressed, partially responded, or fully responded to the indicated treatments.

3. The detection of CSF1 in Fig. 4D seems selective. There are many reasons for tumor-induced spleen enlargement; was there a thorough exclusion process to confirm that CSF1 is the primary cause?

We thank the reviewer for raising this point. We are actively pursuing this line of investigation, and current evidence supports a central role for macrophages in this context. Our data indicate that BP962 tumour cells secrete soluble factors that recruit macrophages and promote epithelial-to-mesenchymal transition (EMT). Although this represents an ongoing and separate study, our preliminary results suggest that CSF1 is among the key cytokines contributing to macrophage recruitment and activation within the tumour microenvironment.

While we have not yet formally excluded all alternative mechanisms underlying tumour-induced spleen enlargement, our current findings point to CSF1-driven macrophage expansion as a potential contributing factor. We have amended our manuscript indicating that additional tumour-related mechanisms may also contribute to spleen enlargement.

4. The abbreviations in the figures, such as SM and A/E, should be defined in the figure legend or in the main text.

All abbreviations used in the figures are now clearly defined in the corresponding figure legends for clarity and consistency.

5. In the analysis of the therapeutic effects of ASTX660/pembrolizumab on the “cold” and “hot” tumor cohorts, it seems that the lack of statistical significance in “cold tumors” cohorts is due to the different number of mice in each cohort.

We thank the reviewer for this valuable comment and have carefully considered the statistical aspects raised. In designing our experiments, we aimed to balance the ethical use of animals with the need for robust and interpretable data. The number of mice per group was initially sufficient to detect large treatment effects; however, given that the observed responses were more modest, we have now increased the cohort size in both control and ASTX/EC/ α PD-1 treatment groups for the BP903 and BP487 models.

In the revised dataset (new Suppl. Fig. 7A-G, see below), a small but statistically significant treatment effect was observed in both tumour types. Consistent with our earlier findings, BP487 tumours exhibited a more pronounced response than BP903 tumours, which remained largely refractory. Importantly, no complete regressions were detected in either cohort, in contrast to BP962, confirming that therapeutic efficacy in these models is inherently limited.

We also evaluated α PD-1 in combination with STING agonist treatment, but this failed to further improve outcomes. Together, these data reinforce that BP903 and BP487 tumours differ intrinsically in their susceptibility to necroptosis-based therapies, consistent with their underlying tumour microenvironmental context.

6. The rigor of the conclusions stated in the article needs to be improved. For example, the sentence "However, animals with less immune infiltrated tumors do not benefit from such treatments, even under necroptotic settings and double immune checkpoint blockade" should be revised. The conclusion should state that there is some benefit, but it is not significant enough. The sentence "First, we evaluated the ability of diABZI to drive STING signaling in BP903 and BP487 organoids in vitro. While treatment with diABZI readily triggered activation of IRF3 and interferon signaling (STAT1 activation) in both organoids, the STING agonist showed little cell death activity in vitro, alone or in combination with IAP antagonism" also needs a more nuanced statement.

We would like to thank the reviewer for highlighting these suggestions. We have modified our ms accordingly.

7. For Fig. 6E and F, including a group with "A/E" would allow for a better comparison of the antitumor effects of the combination of diABZI and "A/E."

We have followed the reviewer's suggestion and have updated our ms to include the A/E treatment group (new Fig. 6D-J).

New Fig. 6D-J. Effect of STING-driven necroptosis on BP487 and BP903 T⁰s. (D) Schematic representation depicting the treatment regimen of BP487 and BP903 tumour bearing-C57BL/6J mice. (E) Tumour growth curves of BP487 T⁰s, treated as indicated. Control (n=12), diABZI (n=9), ASTX660/E (n=7), diABZI/ASTX660 (n=9), diABZI/E (n=10) or diABZI/ASTX660/E (n=6). Each line represents one animal and thick lines represent average tumour growth. Curves represent one independent experiment. (F) Kaplan-Meier survival curves of BP487 tumour-bearing animals, treated as in (D). (G) Pie charts depicting the proportion of mice from (E), which progressed, partially responded, or fully responded to the indicated treatments. (H) Tumour growth curves for BP903 T⁰s, treated as indicated in (D): Control (n = 13), ASTX660/E (n = 6), diABZI/ASTX660 (n = 11), diABZI/ASTX660/E (n = 9). Each line represents one animal and thick lines represent average tumour growth. Curves represent one independent experiment. (I) Kaplan-Meier survival curves of BP903-tumour bearing animals, treated as in (D). (J) Pie charts depicting the proportion of mice from (H), which progressed, partially responded, or fully responded to the indicated treatment.

8. In Fig. 6O, the improvement in antitumor function should be assessed using flow cytometry to detect IFN- γ and perforin, which is better than measuring their mRNA levels.

Following the reviewer's suggestion, we assessed the anti-tumour function of T cells and NK cells by flow cytometry. Our new analysis revealed a marked rise in IFN- γ -producing NK cells and CD8⁺ T cells within the tumour microenvironment following diABZI/ASTX660/E treatment in BP903 tumours. Importantly, we also observed an increased population of polyfunctional CD8⁺ T cells, co-expressing IFN- γ and TNF, indicative of enhanced effector function (new Fig. 6O). Although the frequencies of Perforin- or Granzyme B-expressing cytotoxic cells remained stable (new Suppl. Fig. S9), the data overall point to a more activated and functionally potent anti-tumour immune response induced by diABZI/ASTX660/E treatment.

Reviewer #3 (Remarks to the Author):

Summary of noteworthy results:

This study established tumor organoid lines from Brca1/p53 mouse models of human triple-negative breast cancer and generated orthotopic mammary tumors in immunocompetent mice.

Using this as a platform, they identified one tumor organoid line with both high immune infiltration and high expression of RIPK1, a mediator of apoptosis/necroptosis cell death, and used it to demonstrate the efficacy of RIPK1-induced immunogenic cell death. Specifically, double treatment with a cell-death-inducing agent (ASTX660, an IAP antagonist) and anti-PD1 (pembrolizumab) delayed BP962 tumor progression, whereas triple treatment (adding emricasan, a pan-caspase inhibitor) to promote necroptosis over apoptosis further enhanced the anti-tumor effect, leading to complete cure of the mice and immune memory against subsequent re-inoculation of tumor cells. The treated tumors showed increased T cell and decreased macrophage infiltration. In contrast, the same anti-tumor effect was not seen in tumors induced from two other organoid lines with lower RIPK1 levels and less immune infiltration. The authors therefore proposed RIPK1 as a biomarker for necroptosis-based enhancement of immune checkpoint blockade therapy in immune-infiltrated tumors.

The authors further showed that immune-cold tumors (derived from two organoid lines, BP903 and BP487) can be sensitized to necroptosis-induced cell death (ASTX660/emricasan) by cGAS-STING activation (via diABZI) that activates immune infiltration through viral mimicry. The treatment reshaped the tumor immune microenvironment with more effector lymphocytes and fewer immunosuppressive cells.

Overall assessment:

This study provides interesting and exciting findings that suggest a new potential approach to enhance immunotherapy efficacy for patients with triple-negative breast cancer. This is not the first study to use Brca1/p53-mutant mammary tumor organoids and orthotopic transplantation to study drug response and resistance (see <http://pubmed.ncbi.nlm.nih.gov/29256493/> for example). However, this study used these organoids in an immunocompetent host setting, and the findings of the anti-tumor effect of necroptosis in combination with immune-modulating treatment are novel and promising.

The methodology, analysis, and interpretation are sound. However, more data are needed to strengthen the current conclusion, which is drawn based on the comparison between only one single tumor organoid line (BP962, with high RIPK1 and high immune infiltration) versus two other lines (RIPK-low and/or poor immune infiltration). The experiments should be repeated on additional tumor organoid line(s) with a similar profile to BP962 to establish that the findings are generalizable.

Major comments:

- Does other tumor organoid line(s) with similar RIPK1 expression and immune infiltration to BP962 show the same sensitivity to the combination treatment(s) using ASTX660/pembrolizumab or ASTX660/pembrolizumab/E treatment?

This is an excellent question. Indeed, elevated RIPK1 expression and pronounced immune infiltration may act as indicators of necroptotic responsiveness within the tumour microenvironment.

Both **BP649** and **E0771**-derived tumours exhibit high levels of RIPK1 expression and substantial macrophage infiltration, providing complementary models to investigate this hypothesis.

When establishing BP649 organoids *in vivo*, we observed a longer latency period (>**3 months**) and variable tumour growth dynamics, which prevented us from completing the analysis within the revision timeframe. In parallel, we analysed **E0771-derived tumours**, which also display strong RIPK1 expression and abundant macrophage infiltration, which is highly reminiscent to the one of BP962 tumours (**Suppl. Fig. S6A-G**). Notably, ASTX660/E/ α PD-1 treatment in this model resulted in 100% complete responses.

Together, these findings support the notion that high RIPK1 expression correlates with necroptotic competence and suggest that both tumour-intrinsic factors and the stromal compartment contribute to the therapeutic outcome.

Suppl. Fig. S5. RIPK1 levels. Stacked histogram of RIPK1 protein expression in cancer cells of TPs, ranked in order of their Hscores

New Suppl. Fig. 6B-G. (B) E0771-derived tumours express high levels of RIPK1 and exhibit substantial macrophage infiltration, which are similar to the characteristics of BP962. Representative IHC images of F4/80+ and RIPK1 expression of the indicated tumours. Scale bar = 100 μ m. **(C)** Schematic representation depicting the treatment regimen of E0771 tumour bearing-C57BL/6J mice. **(D)** Tumour growth curves of E0771 derived tumours: Control (n = 8), A/E (n=9) and A/E/ α PD-1 (n=10). Each line represents one animal and thick lines represent average tumour growth. Curves represent two independent experiments. **(E)** Tumour growth kinetics (days 0-38) of mice treated as in (C), measured by the area under the curve (AUC). Each point represents the AUC of individual mice from (D). **(F)** Kaplan-Meier survival curves of E0771 tumour-bearing animals, treated as in (C). **(G)** Pie charts depicting the proportion of mice from (C), which progressed, partially responded, or fully responded to the indicated treatments.

- Is RIPK1 necessary and sufficient for the anti-tumor effects of ASTX660/pembrolizumab or ASTX660/pembrolizumab/E treatment in immune-infiltrated tumors? For example, does knockdown/knockout of RIPK1 in a RIPK1-high tumor organoid line (e.g. BP962) abolish the anti-tumor effects of such combo treatment(s)? Alternatively, can overexpression of RIPK1 in RIPK1-low/immune high tumors enable the anti-tumor effect of such combination treatment(s)?

To investigate the role of RIPK1, we used CRISPR/Cas9 to target *Ripk1* in BP962 (**new Fig. 7A-C and Suppl. Fig. S10**). When *BP962^{RIPK1-KO}* tumour organoids were treated with ASTX660/E, the anti-tumour effect was strongly attenuated, albeit remained statistically significant. However, the additional benefit of α PD-1 co-treatment was lost. This demonstrates that tumour-cell-intrinsic RIPK1 expression is critical for full therapeutic efficacy.

Interestingly, *Ripk1* loss had a more pronounced impact than *Mkl1* deletion in BP962, indicating that RIPK1 fulfils additional signalling roles beyond the execution of necroptosis. These may involve the induction of DAMPs that facilitate cDC1 cross-priming (PMID: 26405229), thereby enhancing adaptive immune activation and responsiveness to α PD-1 therapy.

Together with *BP962^{Mkl1-KO}* and *Mkl1^{-/-}* mouse data (**new Fig. 7D-I**), these findings indicate that necroptotic competence in both tumour and stromal compartments is required for optimal therapeutic response.

New Fig. 7. Cancer cell intrinsic and tumour micro-environmental necroptosis contributes to the anti-tumour effect. (A) Tumour growth curves of BP962 *Ripk1*^{KO} in C57BL/6J mice: Control (n = 8), A/E (n=6) and A/E/αPD-1 (n=9). Each line represents one animal and thick lines represent average tumour growth. Curves represent one independent experiment. (B) Kaplan-Meier survival curves of BP962^{Ripk1-KO} T⁰ tumour-bearing animals. (C) Pie charts depicting the proportion of mice, which progressed, partially responded, or fully responded to the indicated treatments. (D) Schematic representation depicting the experimental setup to evaluate the contribution of tumour versus stromal necroptosis. The treatment regimen is depicted on the right. (E) Tumour growth curves of BP962^{Miki-KO} T⁰s: Control (n = 8), A/E (n = 12) and A/E/αPD-1 (n = 12). Each line represents one animal and thick lines represent average tumour growth. Curves represent one independent experiment. (F) Kaplan-Meier survival curves of BP962^{Miki-KO} tumour-bearing animals, treated as in (D). (G) Pie charts depicting the proportion of mice from (F), which progressed, partially responded, or fully responded to the indicated treatments. (H) Tumour growth curves of BP962 T⁰s grown in *Miki*^{-/-} animals (see Fig. 5E-G for WT control). Treatments: Control (n = 6) and A/E/αPD-1 (n = 9). Each line represents one animal and thick lines represent average tumour growth. Curves represent two independent experiments. (I) Kaplan-Meier survival curves of BP962 tumour-bearing *Miki*-KO animals, treated as in (D). A=ASTX660; E=emricasan.

- What drives the heterogenous expression of RIPK1 in these primary tumours and their derivative tumor organoid lines? Can the authors provide data to suggest a potential mechanism, or at least discuss some speculation in the text?

This is indeed an intriguing question. We observe a clear correlation between macrophage infiltration and elevated *RIPK1* expression. BP962 organoids appear to secrete factors that drive a macrophage-rich tumour microenvironment. We speculate that macrophage-produced cytokines, such as TNF, may stimulate compensatory upregulation of *RIPK1* to protect tumour cells from cell death. While elevated *RIPK1* levels may provide resistance to death ligands under basal conditions, they may also 'prime' these tumours for necroptotic activation upon appropriate triggers (ASTX660/E).

We have expanded our 'Discussion' to discuss this point.

- It would make the paper even stronger if the authors could provide complementary data from humans to reinforce at least some of their conclusions drawn in mouse models.

We fully agree with the reviewer. Indeed, we do have complementary clinical data from our investigator-led **ASTEROID Phase I trial** (see below) that supports our conclusions. However, these data are substantial and are therefore beyond the scope of the current ms and will be reported in a separate ms. Results from our **ASTEROID Phase I trial** (*Targeting IAPs potentiates immune checkpoint blockade in advanced solid tumors*) confirm a similar scenario in humans. In the ASTEROID trial, we combined the IAP antagonist ASTX660 with pembrolizumab (α PD-1) in patients with advanced solid tumours. Excitingly, this combination is not only well tolerated but also showed early signs of efficacy, including long-lasting anti-tumour immune responses in several cancer patients typically resistant to immune checkpoint blockade. This manuscript is currently under evaluation at Nature Medicine.

Minor points:

- “Notably, the individual organoid lines remained stable, and their respective TOs were highly reproducible within each line, as evidenced by mRNA analysis and mutational profiling (Fig. 2e,f)”. Figure 2f showed CNAs present in primary tumors and absent in organoids and vice versa, potentially due to the expansion of clonal subpopulation and/or acquired CNA in culture. The authors should modify their language about stability and reproducibility to reflect these differences between primary tumors and organoid lines.

We agree that some CNAs differ between the primary tumours and their derived organoids, likely reflecting the expansion of specific subclones and/or acquisition of CNAs during culture. We have therefore revised the text to clarify that the term “stable and reproducible” refers to consistency within each organoid line across passages, rather than complete genomic identity with the corresponding primary tumour.

The revised text now reads:

“Notably, the individual organoid lines remained stable and reproducible across passages, as evidenced by mRNA and mutational profiling, while some clonal and CNA differences relative to the parental tumours were observed, reflecting tumour heterogeneity and adaptation in culture (**Fig. 2E,F**).”

- Figure 2h: Scale bars representing different sizes should be more clearly labeled.

We have updated the figures to clearly indicate the differences in the scale bars.

- Figure 3a-b: Are the “normal mammary glands” analyzed after the lymph nodes have been removed? The author should clarify in the figure legend and/or methods section.

We have updated the text in materials and methods.

- Figure 3j: genes should be labeled in the heatmaps.

We have updated Fig 3j to include the genes names

- Figure S5c schematic indicates that cells were stained with PI and Hoechst, whereas the data panels (Fig. 4e,g,h,i, and S5d) show PI and DAPI. Please clarify.

We have corrected this oversight. We have used Hoechst for death assays and DAPI for immunofluorescence.

We would like to thank you and the referees for finding our revised ms much improved.

The referees did not raise any additional comments that needed addressing.

Best wishes

Pascal Meier